# AMPylation matches BiP activity to client protein load in the endoplasmic reticulum

Steffen Preissler[1][*][†], Cláudia Rato[1][†], Ruming Chen[1], Robin Antrobus[1], Shujing Ding[2], Ian M Fearnley[2], David Ron[1][*]

[1]Cambridge Institute for Medical Research, University of Cambridge, Cambridge, United Kingdom; [2]MRC Mitochondrial Biology Unit, Cambridge, United Kingdom

**Abstract** The endoplasmic reticulum (ER)-localized Hsp70 chaperone BiP affects protein folding homeostasis and the response to ER stress. Reversible inactivating covalent modification of BiP is believed to contribute to the balance between chaperones and unfolded ER proteins, but the nature of this modification has so far been hinted at indirectly. We report that deletion of *FICD*, a gene encoding an ER-localized AMPylating enzyme, abolished detectable modification of endogenous BiP enhancing ER buffering of unfolded protein stress in mammalian cells, whilst deregulated FICD activity had the opposite effect. In vitro, FICD AMPylated BiP to completion on a single residue, Thr[518]. AMPylation increased, in a strictly *FICD*-dependent manner, as the flux of proteins entering the ER was attenuated in vivo. In vitro, Thr[518] AMPylation enhanced peptide dissociation from BiP 6-fold and abolished stimulation of ATP hydrolysis by J-domain cofactor. These findings expose the molecular basis for covalent inactivation of BiP.

**\*For correspondence:** sp693@ cam.ac.uk (SP); dr360@medschl. cam.ac.uk (DR)

[†]These authors contributed equally to this work

## Introduction

Protein folding homeostasis in the endoplasmic reticulum (ER) is defended by signal transduction pathways that match the complement of chaperones and enzymes to the burden of unfolded protein within the compartment. Transcriptional activation of genes that enhance the capacity of the ER to process its clients and regulated translation initiation, which controls the flux of unfolded proteins into the ER, comprise the unfolded protein response (UPR) vital to the well-being of cells, tissues and organs (*Balch et al., 2008*; *Walter and Ron, 2011*). Acting alongside this coherent UPR are rapid, activity-dependent post-translational changes in the disposition of the major ER chaperone BiP. Given the dominant role of BiP in protein folding homeostasis in the ER, the latter stands to have considerable biological significance.

The UPR regulates the abundance of BiP transcriptionally (*Chang et al., 1989*; *Kozutsumi et al., 1988*), but this is a latent homeostatic process manifesting over hours and days. On a much shorter time scale BiP's oligomeric state is observed to change, with fewer oligomers present as levels of unfolded protein increase. The architecture of BiP oligomers is consistent with their role as a rapidly accessible repository of inactive BiP that the cell may draw upon in short notice to cope with rapid fluctuations in unfolded protein load (*Preissler et al., 2015*). BiP is also subject to activity-dependent post-translational modification(s). This is reflected in transfer of metabolic label from intracellular pools of tritiated adenosine and [32]P phosphate onto BiP, covalently modifying the protein (*Carlsson and Lazarides, 1983*; *Hendershot et al., 1988*) and imparting upon it a lower isoelectric point (pI) (*Carlsson and Lazarides, 1983*; *Laitusis et al., 1999*). These covalent transformation(s) of BiP correlate inversely with the burden of unfolded proteins in the ER (*Chambers et al., 2012*; *Laitusis et al., 1999*; *Leno and Ledford, 1989*) and because the modified form of BiP was under-represented in complex with substrates (*Hendershot et al., 1988*), were deemed to reflect inactivating modification(s) of the chaperone (*Chambers et al., 2012*).

**eLife digest** Newly made proteins are unstructured chains of amino acids that must fold into particular shapes to work effectively. Proteins that are destined to be exported from the cell undergo folding in a compartment of the cell called the endoplasmic reticulum. This folding is assisted by chaperones (which are themselves also proteins).

Cells adjust the number of chaperones so that there are enough to cope with the burden of unfolded proteins. However, in the endoplasmic reticulum, the known mechanisms that regulate the production of chaperones are too slow to track the rapid fluctuations in the production of unfolded proteins. This suggests that other means exist to balance active chaperones and unfolded proteins that go beyond merely controlling chaperone or unfolded protein abundance.

An important chaperone protein of the endoplasmic reticulum, called BiP, is chemically modified when the production of unfolded proteins declines, and loses the modification when more unfolded proteins are produced. This suggests that the modification might adjust BiP's activity so that it can handle the unfolded proteins that are present. However, previous studies have failed to agree about the nature of the chemical modification and how it affects how BiP works.

Preissler, Rato et al. compared the activity of BiP in normal mammalian cells and in cells engineered to lack an enzyme called FICD. This enzyme attaches a molecule of adenosine mono-phosphate (AMP) to proteins in a process known as AMPylation. The experiments revealed that AMPylation is the modification of BiP that tracks how many unfolded proteins are in the cell. Further studies showed that AMP attaches to a single amino acid of BiP, number 518, a threonine. Reconstructing the AMPylation of threonine 518 in a test tube caused the modified BiP to lose its ability to engage with unfolded proteins.

Overall, Preissler, Rato et al.'s results indicate that cells inactivate BiP by AMPylating threonine 518 as the number of unfolded proteins decreases, and remove the modification to re-activate BiP in response to mounting levels of unfolded proteins. Further studies are now needed to determine how AMPylation inactivates BiP and to understand how the FICD enzyme is regulated so that it performs AMPylation at the right time. It also remains to be explored how important the regulation of BiP activity by AMPylation is for living cells.

Despite considerable effort, the chemical nature of the modification(s) was never directly ascertained. ADP-ribosylation seemed a good candidate as it fits the labeling profile (comprised of adenosine and phosphate), the acidic nature of the modification and its susceptibility to the effect of the ADP-ribosylation inhibitor novobiocin (*Laitusis et al., 1999*). Early work even suggested the presence of an enzymatic activity in cell lysates that could transfer a label from $^{32}P$ $NAD^+$ onto BiP in vitro (*Carlsson and Lazarides, 1983*); but molecular characterization of this enzymatic activity proved elusive. A consensus emerged in regards to the region of BiP that bears the modification, as both $^{32}P$ orthophosphate and $^{3}H$ adenosine labeling mapped consistently to a cyanogen bromide (CnBr) cleavage fragment in the C-terminal substrate binding domain of BiP (Thr$^{434}$ to Met$^{541}$) (*Chambers et al., 2012*; *Gaut, 1997*). Mutations in two arginine residues on that fragment (Arg$^{470}$ and Arg$^{492}$) markedly attenuated radiolabeling in vivo of FLAG-tagged BiP expressed in cells, suggesting that they might be the modification sites (*Chambers et al., 2012*), but peptides with a mass consistent with ADP-ribosylated BiP were never uncovered and the assignments thus remained tentative.

Recently it has been reported that BiP is subject to a different modification, AMPylation, which results in the formation of an phosphodiester bond between the alpha phosphate of ATP and a hydroxyl amino acid side chain (with release of pyrophosphate). This modification of BiP is effected by a broadly conserved, ER-localized enzyme, FICD (also known as HYPE) (*Ham et al., 2014*; *Sanyal et al., 2015*). There is lack of unanimity in regards to the circumstances and consequences of BiP AMPylation: the Orth lab provides evidence that AMPylation is an inactivation modification, induced when client protein burden is low (*Ham et al., 2014*), whereas the Mattoo lab suggests that it is an activating modification induced by ER stress (*Sanyal et al., 2015*). AMPylation can explain both the metabolic labeling profile of BiP (from adenosine and phosphate pools) as well as the

acidity of the modified form and possibly even the sensitivity to novobiocin (an ATP mimetic). However, the assignment of AMPylation to Ser[365] or Thr[366] in the nucleotide binding domain of BiP (*Ham et al., 2014*; *Sanyal et al., 2015*) is at odds with the cyanogen bromide cleavage pattern of metabolically labeled BiP, suggesting a complex scenario of multiple modifications with alternative outcomes.

Here we report on a functional and quantitative analysis of BiP modification by FICD in vitro and in vivo. Our findings fit best a parsimonious model whereby AMPylation of BiP on a single residue, Thr[518], is the only quantitatively significant modification of the chaperone and provide clues to its functional significance.

## Results

### FICD is necessary and sufficient for conversion of endogenous BiP to an acidic modified form with altered mobility on native-PAGE

Changes in the disposition of BiP induced by manipulation of conditions in the ER are conveniently tracked by native gel electrophoresis (native-PAGE) and immunoblotting (*Freiden et al., 1992*) coupled with site-directed proteolysis (*Preissler et al., 2015*) (*Figure 1A*). Inhibition of protein synthesis led to accumulation of a prominent high mobility species tentatively named 'B' form, whereas depletion of ER calcium progressively drew on this pool to promote the assembly of BiP oligomers (*Preissler et al., 2015*) (*Figure 1B and C*). The 'B' form of BiP was also noted for its relative resistance to cleavage in vitro by a bacterial protease, SubA, which cuts BiP within a highly conserved linker sequence connecting the nucleotide binding domain (NBD) and substrate binding domain (SBD) (*Paton et al., 2006*) (*Figure 1A and D*). Resistance to cleavage by SubA is also a feature of the acidic, modified form of BiP (*Chambers et al., 2012*), suggesting that the 'B' form observed on native-PAGE might be related to the covalently modified form of BiP (previously believed to be the ADP-ribosylated form).

To examine the potential role of FICD in elaboration of the 'B' form of BiP, we inactivated both alleles of *FICD* in hamster CHO-K1 cells by CRISPR-Cas9-mediated genome editing, truncating the coding sequence N-terminal to the FIC-domain active site (*Figure 2A*). The SubA-resistant 'B' form, prominent in cycloheximide-treated wildtype cells, was conspicuously absent in *FICD*[-/-] cells (compare lanes 2 and 5 in *Figure 2B*), whilst other aspects of BiP's structural dynamism (such as oligomerization in response to ER calcium depletion) remained largely unchanged. *FICD* inactivation also led to disappearance of the acidic form of BiP, present in lysates of cycloheximide-treated wildtype cells resolved by isoelectric focusing PAGE (IEF-PAGE, *Figure 2C*). FICD-mediated AMPylation is strongly autoinhibited by intramolecular competition for ATP γ-phosphate binding by glutamate 234, whose mutation to glycine (FICD[E234G]) de-represses FICD, whereas replacement of histidine 363, which is located in the highly conserved catalytic FIC motif, with alanine (FICD[H363A]) abolishes AMPylation activity (*Engel et al., 2012*). In keeping with these observations, enforced overexpression of the constitutively active FICD[E234G] led to the re-appearance of an acidic form of BiP in the *FICD*[-/-] cells (*Figure 2D*). Wildtype FICD did not promote BiP's acidic form, even when strongly overexpressed, suggesting the existence of robust inhibitory mechanisms restraining potentially deleterious effects of protein modification (*Engel et al., 2012*). Such tight regulation is a feature common to FIC enzymes (*Garcia-Pino et al., 2014*) but the signals leading to derepression under physiological circumstances are unknown so far.

The *FICD*[-/-] cells provided a convenient experimental reference against which to gage time dependent changes in the abundance of modified BiP following the imposition of ER stress and during its resolution in wildtype cells. A decline in 'B' form and an increase in susceptibility to digestion by SubA were noted within 2 hr of exposure of cells to the reversible ER stress inducing agent 2-deoxyglucose (*Figure 2—figure supplement 1A and B*) and persisted for up to 8 hr thereafter. This decline in 'B' form occurred in the face of a marked increase in *FICD* mRNA (*Figure 2—figure supplement 1D*) (*Ham et al., 2014*; *Sanyal et al., 2015*), a finding consistent with regulatory mechanisms that contravene the increase in mRNA. Washout of 2-deoxyglucose was associated with progressive increase in 'B' form and the emergence of resistance to cleavage by SubA (*Figure 2—figure supplement 1A and C*).

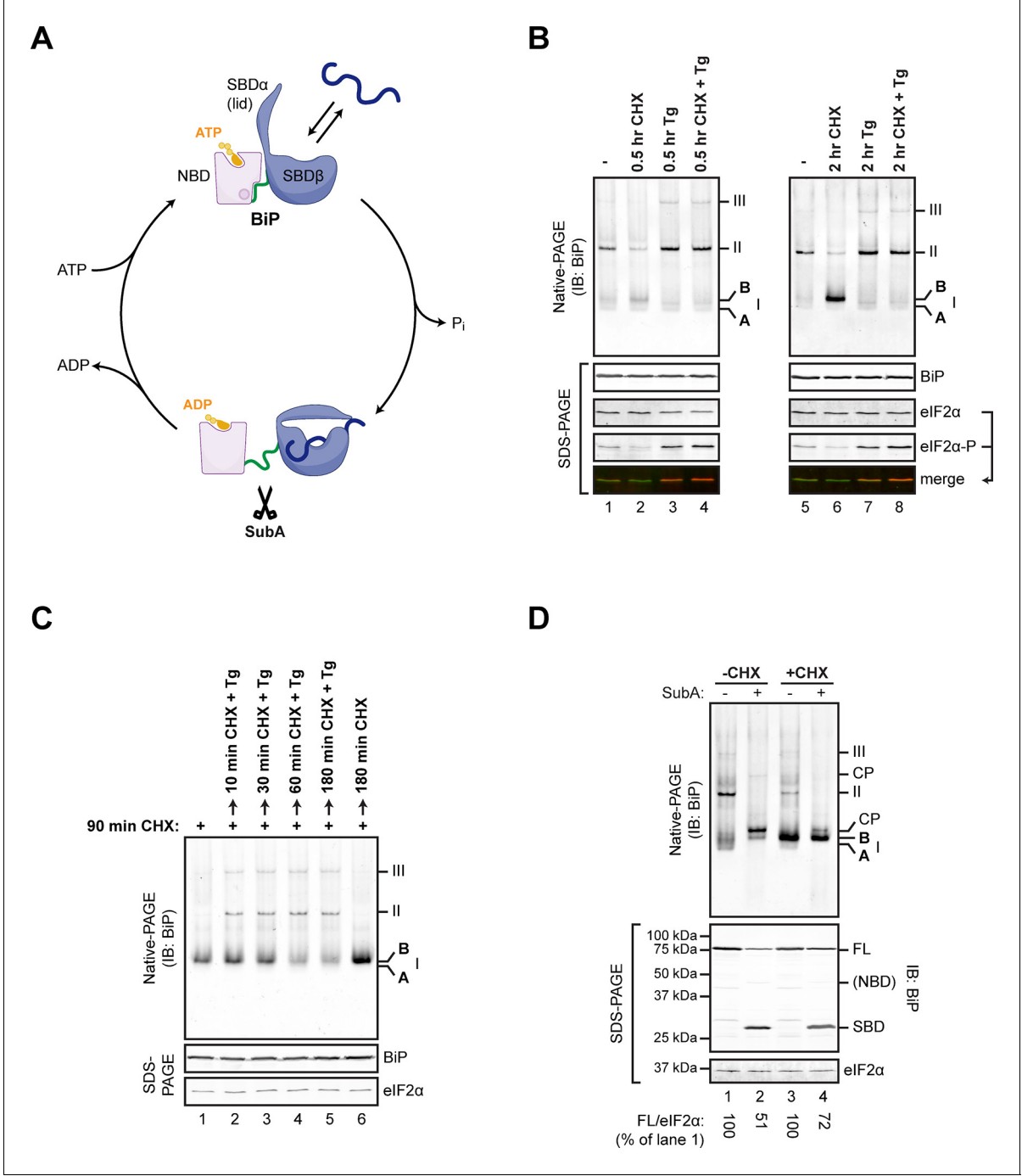

**Figure 1.** Native gel electrophoresis tracks activity state-dependent changes in BiP's quaternary structure. (**A**) Schematic representation of BiP's domain organization in the ATP- and ADP-bound states. BiP consists of an N-terminal nucleotide binding domain (NBD, pink) and a C-terminal substrate binding domain (SBD, blue) connected by a hydrophobic interdomain linker peptide (green). The SBD is composed of a two-layered β-sandwich subdomain (SBDβ) containing the substrate binding crevice and a helical lid structure (SBDα). In the ATP-bound conformation the NBD and SBD form extensive contacts, which involves the linker region, and the SBD is in the open conformation (SBDα extended) allowing for interactions with substrates (dark blue) at high association and dissociation rates. Upon ATP hydrolysis to ADP the inter-subunit contacts are lost leading to exposure of the linker, packing of SBDα against SBDβ, and strong reduction of substrate interaction kinetics. Cleavage of BiP by the linker-specific protease SubA (scissors) is favored in the ADP-bound state. (**B**) Immunoblot of endogenous BiP resolved by native gel electrophoresis. Where indicated the CHO-K1 cells were exposed to cycloheximide (CHX; 100 μg/mL) or thapsigargin (Tg; 0.5 μM) for the indicated time. The major species visible on the native gel are numbered by order of descending mobility (I-III) and the monomeric 'B' form induced by CHX treatment and the 'A' form detectable in untreated cells are marked. Immunoblots of the same samples resolved by SDS-PAGE report on total BiP and total eIF2α (which also serves as a loading control) and phosphorylated eIF2α to reveal the impact of thapsigargin action. Note the emergence of the 'B' form in CHX-treated cells, which is blocked by

*Figure 1 continued on next page*

Figure 1 continued

thapsigargin. (C) BiP immunoblot, as in "A". Cells were first exposed to cycloheximide to build a pool of the 'B' form of BiP and then challenged with thapsigargin (in the continued presence of cycloheximide). Note the disappearance of the 'B' form of BiP and the emergence of BiP oligomers in the thapsigargin-treated cells. (D) BiP immunoblot of lysates from untreated (-CHX) and cycloheximide-treated (+CHX) cells. Where indicated the lysates were exposed to the SubA protease that cleaves BiP's interdomain linker in vitro, before loading onto the native gel. The cleavage products detected by the antiserum used on the native gel are noted (CP, upper panel), as are the full-length BiP (FL) and its substrate binding domain (SBD) on the SDS-PAGE gel below (the nucleotide binding domain is very weakly reactive with the antiserum). eIF2α serves as a loading control. Note the resistance of the 'B' form of BiP to cleavage by SubA.

These changes fit well the previously observed inverse correlation between levels of ER stress and the abundance of modified acidic BiP (*Chambers et al., 2012*; *Laitusis et al., 1999*). Together with evidence that FICD was both necessary and sufficient for elaboration of the acidic, modified form of BiP on IEF-PAGE, these observations lent strong support to the notion that the 'B' form of BiP, observed on native gels, reflects the same or a related species. The high mobility of the FICD-dependent 'B' form and its indifference to the effects of ATP (*Figure 2B*, lower panel) (which promotes dissociation of BiP from its client proteins in vitro) are consistent with a role for the FICD-mediated modification in BiP inactivation. To examine these relationships in further detail we measured the effects of FICD on BiP in vitro, in a system constituted of pure components.

Whether purified as a GST-fusion protein from bacteria or from overexpressing mammalian cells, active FICD$^{E234G}$ promoted the ATP-dependent in vitro appearance of a 'B' form of recombinant BiP purified from bacteria. Like the endogenous 'B' form, found in cells, the 'B' form constituted in vitro was also partially resistant to cleavage by SubA (*Figure 3A*). Emergence of the 'B' form in vitro correlated with the acquisition of a faster migrating acidic form of BiP on IEF-PAGE (*Figure 3B*). FICD converted all the detectable BiP to a single acidic form in a time-dependent manner, which, like the endogenous acidic from of BiP, was also relatively resistant to cleavage by SubA (*Figure 3B*, lane 8). Likewise, addition of purified active FICD$^{-/-}$ to lysates from *FICD$^{-/-}$* cells entirely converted endogenous BiP into the acidic form (*Figure 3C*). Of note, within the resolution of IEF-PAGE, there is no evidence for heterogeneity in BiP modification [also see (*Carlsson and Lazarides, 1983*; *Laitusis et al., 1999*)]. This observation is consistent either with processive modification of BiP or modification occurring on a single site in any given BiP molecule.

## FICD AMPylates BiP on Thr$^{518}$ in vitro

FICD efficiently transferred the $^{32}$P label from the alpha phosphate of ATP onto BiP, consistent with AMPylation (*Figure 3D*). Radiolabeled BiP too was resistant to cleavage by SubA indicating that BiP radiolabeled in vitro by FICD$^{E234G}$ reports faithfully on the behavior of endogenous BiP (*Figure 3—figure supplement 1*). Interestingly, cleavage of radiolabeled BiP (by prolonged incubation with SubA) led to emergence of a radiolabeled substrate binding domain fragment (Leu$^{417}$-Leu$^{654}$); the nucleotide binding domain fragment, though conspicuous on the Coomassie-stained gel, was entirely devoid of label (*Figure 3D* and *Figure 3—figure supplement 1B*). This in vitro labeling pattern fits well with the metabolic labeling of BiP by $^{32}$P orthophosphate or $^{3}$H adenosine label donors in vivo, as the label was mapped to a single CnBr cleavage fragment spanning Thr$^{434}$ to Met$^{541}$ (*Chambers et al., 2012*; *Gaut, 1997*). In contrast, FICD-mediated transfer of $^{32}$P label from ATP to BiP's substrate binding domain, observed here in vitro, can not be reconciled with reported AMPylation of Thr$^{366}$ or Ser$^{365}$ (*Ham et al., 2014*; *Sanyal et al., 2015*).

The migration of purified recombinant BiP on IEF and native gels suggests that most or all of the protein was modified in vitro with FICD$^{E234G}$, yielding homogenous preparations of modified BiP, whereas BiP treated with the inactive mutant FICD$^{E234G-H363A}$ remained unmodified (*Figure 3A and 3B*). These preparations were examined by electrospray ionization mass spectrometry and produced molecular masses of 71,060.1 and 71,395.6 Da for unmodified and modified BiP, respectively (*Figure 4A* and *Figure 4—figure supplement 1*). These two species were the only proteinaceous ions detected in the samples; the trailing shoulder of the ion-current traces (*Figure 4—figure supplement 1A*) being comprised entirely of singly charged low molecular mass contaminants. The mass difference of 335 Da is consistent with a single covalent modification by AMP (329.12 Da) in modified BiP. The difference between the predicted mass of a molecule of AMP bound via an ester

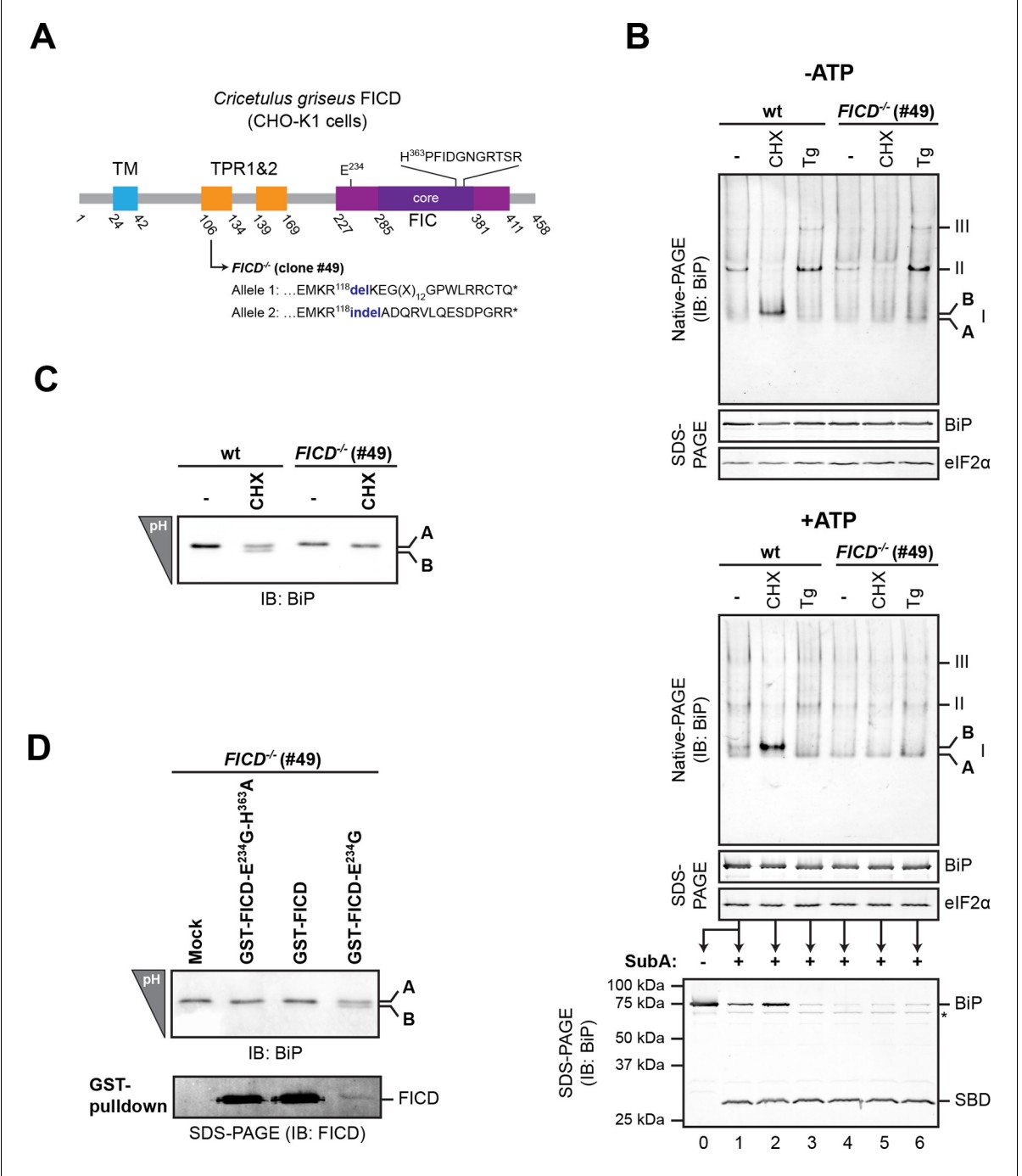

**Figure 2.** *FICD* deletion abolishes BiP modification in cultured cells. (**A**) Schematic illustration of the hamster FICD protein. The transmembrane region (TM), the two tetratricopeptide repeats (TPR) as well as the FIC-domain (purple) with its core sub-domain (dark purple) and the catalytic loop sequence are shown. Numbers represent amino acid positions. The amino acid sequence surrounding the mutations introduced into the CHO-K1 *FICD*⁻/⁻ clone (#49) by CRISPR-Cas9-mediated genome editing are noted. Both alleles result in premature termination of translation deleting the active site (*). (**B**) Immunoblots of endogenous BiP from wildtype (wt) or *FICD*⁻/⁻ CHO-K1 cell lysates from which ATP was either depleted by incubation with hexokinase and glucose (-ATP, top panel) or to which ATP (1 mM) had been added (+ATP, bottom panel), resolved by native-PAGE. Where indicated the cells were exposed to cycloheximide (CHX, 100 µg/ml) or thapsigargin (Tg, 0.5 µM) for 3 hr before lysis. The major species visible on the native gels are numbered by order of descending mobility (I-III) and the monomeric 'B' form induced by CHX treatment and the 'A' form, prominent in ATP-replete lysates of untreated cells, are marked. Immunoblots of the same samples resolved by SDS-PAGE report on total BiP loaded and on eIF2α as a loading control. The ATP-supplemented lysates (3 µg/µl protein) were in addition exposed to SubA (30 ng/µl) for 10 min at room temperature prior to separation by SDS-PAGE and immunodetection of BiP. The intact protein and the substrate binding domain (SBD), which are detected by the antibodies against a C-

Figure 2 continued

terminal epitope of BiP, are indicated. The asterisk marks a band of unknown identity. Note that neither CHX-dependent conversion of endogenous BiP into the monomeric 'B' form nor the CHX-mediated resistance of BiP towards proteolytic cleavage by SubA, were observed in *FICD*[-/-] cells. (**C**) Immunoblot of endogenous BiP from wildtype and *FICD*[-/-] CHO-K1 cell lysates resolved on an isoelectric focusing (IEF) gel. Where indicated the cells have been exposed to CHX (100 μg/ml) for 3 hr before lysis. Note that the more acidic ('B') form of BiP associated with CHX treatment was absent in *FICD*[-/-] cells. (**D**) IEF immunoblot of endogenous BiP from CHO-K1 *FICD*[-/-] cells transfected with plasmids encoding wildtype GST-FICD, the constitutively active GST-FICD[E234G] or the inactive GST-FICD[E234G-H363A] mutant. Mock transfected cells were analyzed as a control. The cells were treated with CHX (100 μg/ml) for 3 hr before lysis. A pulldown with GSH-Sepharose beads was performed with the same lysates to analyze expression levels of the plasmid-encoded GST-FICD fusion proteins. Note that formation of the acidic ('B') form of BiP was restored by expression of catalytically active GST-FICD[E234G] protein (despite its comparatively low expression level) but neither by expression of the catalytically inactive GST-FICD[E234G-H363A] mutant nor the regulated wildtype enzyme.

The following figure supplement is available for figure 2:

**Figure supplement 1.** Time-dependent changes in BiP abundance, BiP 'B' form and the fraction of BiP resistant to cleavage by SubA in cells exposed to the reversible ER stress-inducing agent 2-deoxy-D-glucose (2-DG).

linkage (329.12 Da) and the observed mass difference between the modified and unmodified BiP (335 Da) is reflective of the broad peaks associated with these measurements. These observations indicate a single quantitative modification of BiP by AMPylation and also fit the simple pattern of modified BiP migration on the IEF-PAGE gel. An analogous mass spectrometry experiment with endogenous BiP immunopurified from cycloheximide-treated wildtype CHO-K1 cells also revealed only two molecular masses (70,538.3 Da and 70,866.1 Da) with a mass difference of 327.8 Da, consistent with modification of BiP by a single AMP moiety in vivo. BiP from AMPylation-deficient *FICD*[-/-] cells gave rise to a single species with a molecular mass (70,532.2 Da) that is similar to the predicted mass of unmodified mature BiP (*Figure 4—figure supplement 2*).

To identify the AMPylation site(s), samples of unmodified BiP or BiP modified in vitro with FICD[E234G] were treated with several proteases and the peptide digests were analyzed by liquid chromatography and tandem mass spectrometry (LC-MS/MS). A unique peptide (doubly charged peak of 1374.15 m/z) corresponding in mass to an AMPylation of residues 511-532 was observed in samples digested with Arg-C, and another peptide (doubly charged peak of 916.94 m/z) corresponding to AMPylation of residues 515-528 was detected in the Asp-N digest (*Figure 4B and C*). Moreover, another doubly charged peak (1209.62 m/z) corresponding to the unmodified BiP 511-532 fragment was present in the Arg-C digests of unmodified BiP and absent from the spectrum of the modified form (*Figure 4C*, left panels) enhancing the confidence in these assignments.

No masses corresponding to peptides overlapping BiP 511-532 [the region identified in the Arg-C and Asp-N digests to contain AMPylation site(s)], were detected in either the chymotrypsin or trypsin digests of AMPylated BiP. However, all four digests contained peptides covering Ser[365] and Thr[366], but these were all unmodified peptides (*Figure 4—figure supplement 3*). In contrast to the unmodified 511–532 Arg-C fragment that was absent from the spectrum of the modified BiP (*Figure 4C*), in all four digests the relative intensity of the signals from peptides encompassing Ser[365] and Thr[366] was undiminished by modification with FICD[E234G] (*Figure 4—figure supplement 3B*); this despite evidence that all the detectable BiP molecules in the sample were modified (*Figure 3B* and *4*). The mass spectrometry-based evidence for non-modification of Ser[365] or Thr[366] also fits with the observation that under the in vitro conditions studied here, FICD[E234G] (using α-[32]P-ATP as a substrate) selectively modified a BiP fragment C-terminal to the SubA cleavage site (at Leu[416]) with no evidence for modification of the N-terminal nucleotide binding domain (*Figure 3D* and *Figure 3—figure supplement 1*).

The LC-MS/MS fragment spectra of AMP modified peptides from Arg-C and Asp-N digests obtained by high energy collisional induced dissociation (HCD) or electron transfer dissociation (ETD) did not directly identify the modified residue(s). However, the specificity of FICD for AMPylation of hydroxyl side chains and the overlap of modified peptides from the two digests narrows the modification site to three residues: Thr[518], Thr[525] and Thr[527]. Both BiP[T525A] and BiP[T527A] remained substrates for AMPylation in vitro, whereas the BiP[T518A] mutation abolished all modification of BiP (*Figure 5A and B*).

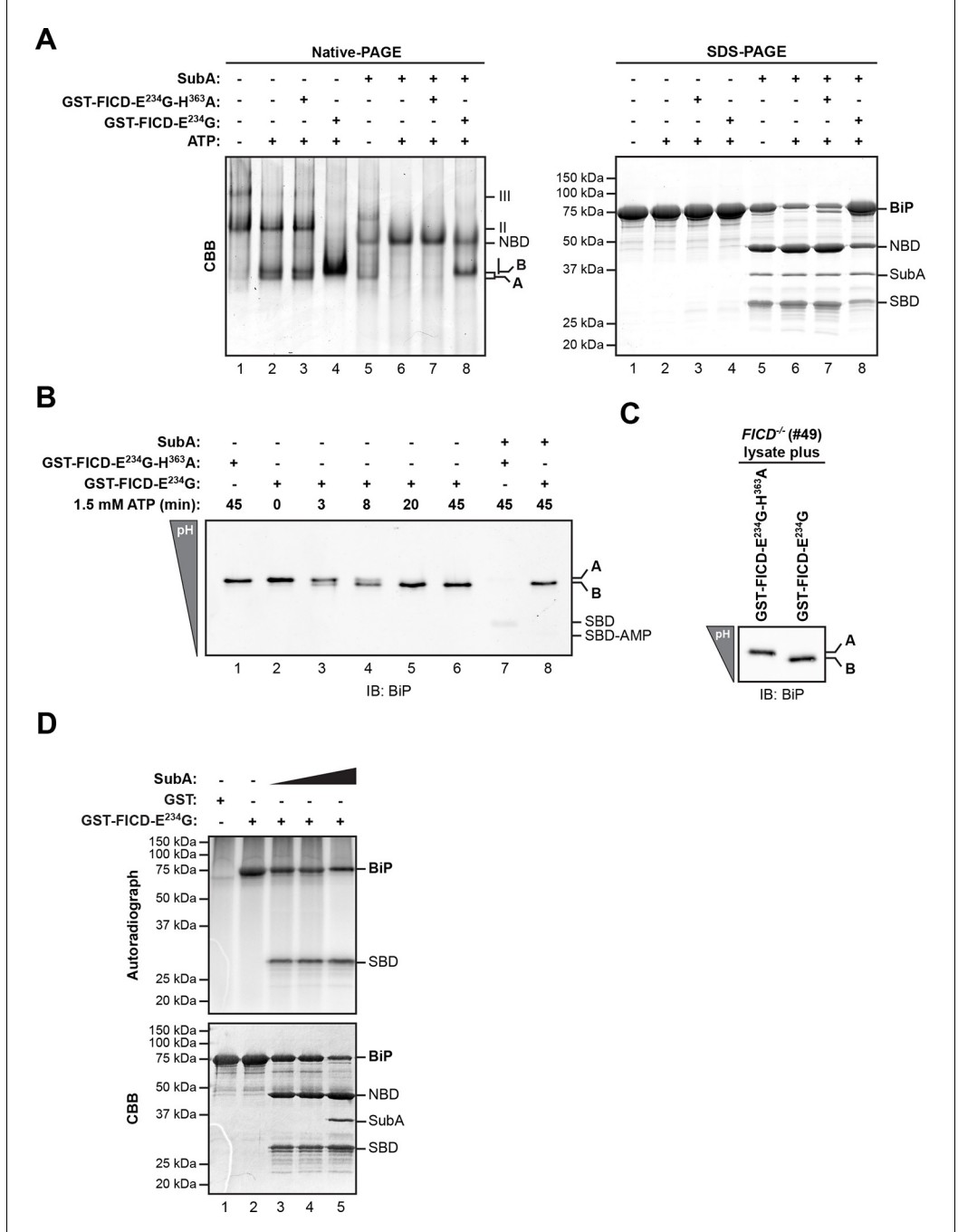

**Figure 3.** AMPylation of purified BiP in vitro recapitulates features of BiP modified in vivo. (**A**) Coomassie (CBB)-stained native-PAGE gel (left panel) or SDS-PAGE gel (right panel) of recombinant BiP purified from bacteria (10 µM) exposed to ATP (1.5 mM) in the absence or presence of recombinant active GST-FICD$^{E234G}$ or inactive GST-FICD$^{E234G-H363A}$ (0.5 µM) purified from *E. coli* (45 min at 30°C). The interdomain linker-specific protease SubA (30 ng/µl) was added afterwards for 10 min where indicated. The major species on the native gel are numbered by order of descending mobility (I-III) and the monomeric 'A' and 'B' forms of BiP are indicated. Full-length BiP, the nucleotide binding domain (NBD, also resolved on the native gel) and the substrate binding domain (SBD) are annotated on the SDS-PAGE gel. Note the quantitative AMPylation-dependent conversion of BiP into the monomeric 'B' form on native gels and the increased resistance of modified BiP to cleavage by SubA. Also note that upon incubation with ATP unmodified BiP forms a second slower migrating monomeric species similar to the 'B' form, which likely reflects an alternative (e.g. ATP-bound) conformation. (**B**) Immunoblot of an IEF gel of purified hamster BiP$^{19-654}$ (15 µM) after in vitro AMPylation with GST-FICD$^{E234G}$ or as a control with inactive GST-FICD$^{E234G-H363A}$ (both at 0.75 µM) in presence of ATP (1.5 mM) for the indicated times at 30°C followed by treatment with or without SubA (30 ng/µl) for 10 min. The two forms of full-length BiP ('A' and the more acidic 'B' form) as well as faint signals that likely represent the unmodified and modified SBD are indicated. (**C**) Immunoblot of endogenous BiP from lysates of cycloheximide-treated CHO-K1 *FICD*$^{-/-}$ cells resolved by IEF-PAGE

*Figure 3 continued on next page*

*Figure 3 continued*

following exposure in vitro to purified active or inactive FICD (as in "B" above). Note the conversion of all the detectable BiP to an acidic form in the sample exposed to active FICD. (D) Autoradiograph and Coomassie stain of an SDS-PAGE gel of BiP exposed in vitro to active GST-FICD$^{E234G}$ coupled to GSH-Sepharose beads (or GST alone as a control) in the presence of α-$^{32}$P-ATP as a substrate. After the AMPylation reaction the samples were treated further with increasing concentrations of SubA (0.03 ng/µl, 0.1 ng/µl and 20 ng/µl, lanes 3-5) where indicated. Note the confinement of the radiolabel to the SBD fragment of cleaved BiP.

The following figure supplement is available for figure 3:

**Figure supplement 1.** Comparison of the differential susceptibility of unmodified and AMPylated BiP to cleavage by SubA in vitro.

BiP$^{T366A}$ and BiP$^{R470E}$ or BiP$^{R492E}$ [the latter are mutations that attenuate labeling of FLAG-tagged BiP in vivo by $^{32}$P orthophosphate or $^3$H adenosine (*Chambers et al., 2012*)] were all modified by FICD$^{E234G}$ in vitro with varying efficiency, as was the substrate binding-deficient mutant BiP$^{V461F}$ (*Figure 5C*). The substitution of Thr$^{518}$ by either alanine or glutamic acid moderately impaired the ability of BiP to form oligomers and slightly modified the mobility of the monomeric ('A') form of BiP on native gels, whereas mutating Thr$^{366}$ had no effect on oligomerization (*Figure 5D*). BiP$^{T518E}$ could not be AMPylated in vitro (*Figure 5C*). Yet, whether exposed to FICD$^{E234G}$ or not, BiP$^{T518E}$ resembled AMPylated BiP by its migration on native-PAGE and by its resistance to SubA-mediated proteolytic cleavage (*Figure 5D*). These observations are consistent with the bulky and negatively charged side chain of glutamic acid mimicking some aspect of AMPylation at Thr$^{518}$, promoting a linker-protected conformation. Together these observations support Thr$^{518}$ as the only FICD-mediated AMPylation site in vitro and suggest that changes at Thr$^{518}$ introduced by AMPylation (or mutation) have significant structural and functional consequences.

## FICD AMPylates BiP on Thr$^{518}$ in vivo

To evaluate sites of FICD-mediated BiP modification in vivo, we took advantage of a CHO-K1 cell line in which the *FICD* gene had been inactivated by CRISPR-Cas9-mediated gene editing (*Figure 2*). Wildtype and *FICD* knockout CHO-K1 cells were cultured in media containing arginine with stable isotopes of either "light" or "heavy" nitrogen and carbon (resulting in a net mass difference of 10 Da per arginine residue) and exposed to cycloheximide (to enhance FICD-dependent modification of BiP). Endogenous BiP was recovered by immunoaffinity purification, digested with proteases and the resultant peptides were subjected to LC-MS/MS analysis. The SILAC (<u>s</u>table <u>i</u>sotope <u>l</u>abeling with <u>a</u>mino acids in <u>c</u>ell <u>c</u>ulture) procedure was designed to quantify relative differences in the abundance of unmodified and modified species in differentially labeled cultures of cells (*Figure 6A*).

Peptides from Arg-C digests corresponding in mass to the unmodified and AMPylated BiP$^{511-532}$ were noted in spectra derived from both untreated and cycloheximide-treated wildtype cells. The unmodified species was 1.5-fold more abundant in the untreated sample, whereas the AMPylated species was 1.5-fold more abundant in the cycloheximide-treated sample (*Figure 6B*). Unmodified BiP$^{511-532}$ was 2.5-fold more abundant in the cycloheximide-treated *FICD* knockout sample than in the cycloheximide-treated wildtype sample, whereas no modified BiP$^{511-532}$ was detected in the cycloheximide-treated *FICD* knockout sample (*Figure 6C*). These figures attest to a large fraction of AMPylated BiP in CHO-K1 cells growing in SILAC media (especially when compared to cells growing in complete media, *Figure 2C*); perhaps a reflection of depletion wrought by dialysis of the serum.

The region encompassing Ser$^{365}$ and Thr$^{366}$ was well represented by peptides from the Arg-C digest. The quantification data showed no change in the abundance of the unmodified Arg-C peptide encompassing Ser$^{365}$ and Thr$^{366}$ (BiP$^{337-367}$, *Figure 6—figure supplement 1*). The contrast between the consistent cycloheximide-dependent and *FICD* genotype-dependent variation in abundance of the peptide containing Thr$^{518}$ and absence of change in the abundance of peptides containing Ser$^{365}$ and Thr$^{366}$, indicates that the latter residues are unlikely to be major sites of BiP AMPylation in CHO-K1 cells.

The mass spectra of the intact protein indicates that FICD modifies any given BiP molecule on a single site both in vitro and in vivo (*Figure 4* and figure supplements) and the peptide fragment analysis indicates that the modification site(s) are all encompassed by peptides spanning 511-532 (*Figure 6* and figure supplements). The fragmentation spectra of Arg-C peptides 511-532 from both

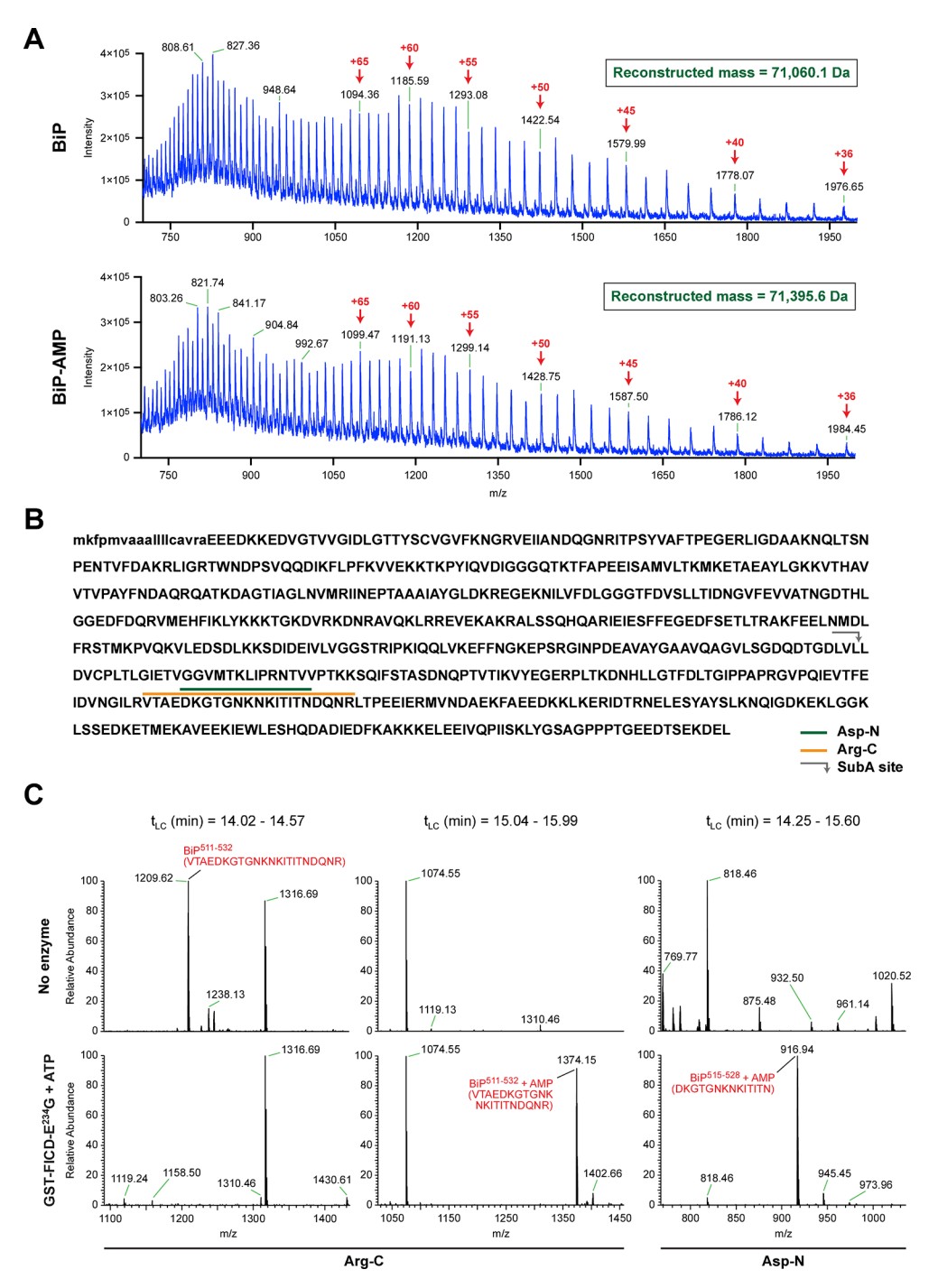

**Figure 4.** FICD-mediated incorporation of a single AMP molecule onto the substrate binding domain of BiP in vitro. (**A**) Electrospray mass spectra of bacterially expressed hamster BiP (27-654, with a His6-tag) after reverse-phase HPLC purification. The spectra contain protein ions with between 36 and 100 associated protons (the number of protons are indicated for the major species). The inset shows the data reconstructed onto a true mass scale. The sample in top panel is of unmodified BiP and that in bottom panel of BiP that had been modified in vitro with GST-FICD$^{E234G}$ and ATP. (**B**) Amino acid sequence of Chinese hamster BiP (with the cleaved signal peptide in lower case letters). The SubA cleavage site (after L416) is marked by the grey arrow and the predicted Arg-C and Asp-N AMPylated proteolytic cleavage fragments are delineated by the colour coded horizontal lines above the protein sequence. (**C**) LC-MS spectra of peptides derived from recombinant BiP digested with Arg-C or Asp-N before ("No enzyme") and after in vitro modification with GST-FICD$^{E234G}$ and ATP. The m/z ratio of the signals is displayed in the abscissa and their relative intensity in the ordinate. The interval of the liquid chromatogram at which the peptides in question eluted is depicted above each paired sample [$t_{LC}$ (min)]. Note the absence of any signal corresponding to the doubly-charged non-AMPylated 511-532 Arg-C fragment in the spectrum derived from BiP after exposure to GST-

*Figure 4 continued on next page*

*Figure 4 continued*

FICD[E234G] and ATP (left-most lower panel) and the absence of signals corresponding to the doubly-charged AMP modified peptides in the spectra derived from the Arg-C or Asp-N digests of BiP that had not been exposed to GST-FICD[E234G] and ATP (central and rightmost upper panels).

The following figure supplements are available for figure 4:

**Figure supplement 1.** Chromatographic profile and reconstructed mass spectrum of unmodified BiP and BiP modified in vitro with FICD and ATP.

**Figure supplement 2.** Modification of BiP with a single AMP molecule in vivo.

**Figure supplement 3.** Evidence for the absence of modification of Ser[365] or Thr[366] in mono-AMPylated BiP.

endogenous BiP AMPylated in vivo, or recombinant BiP AMPylated in vitro, yielded a collection of *y*-ions and *b*-ions that included all the residues on the 511-532 peptide except Thr[518], whereas the latter residue was conspicuously represented in the ion series from the fragmentation spectrum of unmodified endogenous and recombinant BiP (*Figure 6—figure supplement 2*). The HCD procedure likely fragments the modified peptide in an unpredictable manner, rendering it unrecognizable, however, in relief, the fragmentation pattern reveals Thr[518] as the only modification site of BiP, in vitro and in vivo.

## AMPylation efficiency depends on BiP's conformation

In vitro radiolabeling revealed that amino acid substitutions distant from the primary AMPylation site (Thr[518]) notably influenced modification efficiency. Such point mutations might have indirectly influenced the accessibility of Thr[518] to FICD by altering the ability of BiP proteins to interact with other molecules or by changing their overall conformation. In particular, we observed stronger modification of BiP[V461F] and BiP[T366A] compared to wildtype BiP (*Figure 5C*). The V461F mutation may enhance access of FICD to BiP simply by reducing competing oligomerization interactions amongst BiP proteins. In contrast, Thr[366] is located in the nucleotide binding domain and mutations at this position may interfere with ATP binding or hydrolysis and thus affect the conformation of BiP. The relationship between BiP's conformation and the efficiency with which it is modified by FICD was systematically tested by analysis of a series of well-characterized BiP mutants that favor certain conformational states based on their altered abilities to interact with and hydrolyze nucleotides (*Gaut and Hendershot, 1993*; *Petrova et al., 2008*; *Wei et al., 1995*).

BiP[E201G] and BiP[T229A], mutants that bind ATP and undergo allosteric transitions but are defective in nucleotide hydrolysis (*Gaut and Hendershot, 1993*; *Petrova et al., 2008*; *Wei et al., 1995*), exhibited enhanced AMPylation rates compared to wildtype BiP (*Figure 7A*). By contrast, BiP[T37G], which is defective in ATP hydrolysis and in the allosteric transitions upon ATP binding, showed slightly reduced AMPylation, whereas mutants that are severely defective in adopting the ATP-bound conformation, such as the ATP binding-deficient BiP[G226D] or BiP[ADDA], which is locked in the domain-uncoupled state due to a four residue substitution in its interdomain linker (*Laufen et al., 1999*; *Preissler et al., 2015*), remained unmodified by FICD (*Figure 7A and B*). Consistent with an important role for the conformation of BiP in determining its ability to serve as a substrate for FICD, the isolated BiP substrate binding domain (purified from bacteria as a fusion to yeast Smt3) was not modified by FICD[E234G] (*Figure 7—figure supplement 1*).

Kinetic differences in modification rates of mutant BiP molecules, suggested by the radiolabeling experiments, were further confirmed by tracking the time-dependent formation of the 'B' form of BiP following exposure to FICD[E234G] (*Figure 7C*). The $t_{1/2max}$ values for in vitro AMPylation of BiP[T229A] (4.0 min) and BiP[E201G] (2.3 min) were significantly shorter than for wildtype BiP (9.9 min) (*Figure 7D*). ATP is a substrate for FICD, nonetheless, these important kinetic differences in modification were unlikely to reflect the trivial consequence of varying contributions of ATP hydrolysis (by BiP) to the availability of substrate nucleotide for the AMPylation reaction. Accelerated modification was not shared by the ATP hydrolysis-defective BiP[T37G]. Furthermore, BiP AMPylation was performed in the presence of millimolar ATP, whereas the affinity of FICD[E234G] for nucleotide is measured in the 100 nM range (*Bunney et al., 2014*). Together, these observations indicate that modification of BiP is modulated by its conformation and BiP in the compact ATP-bound

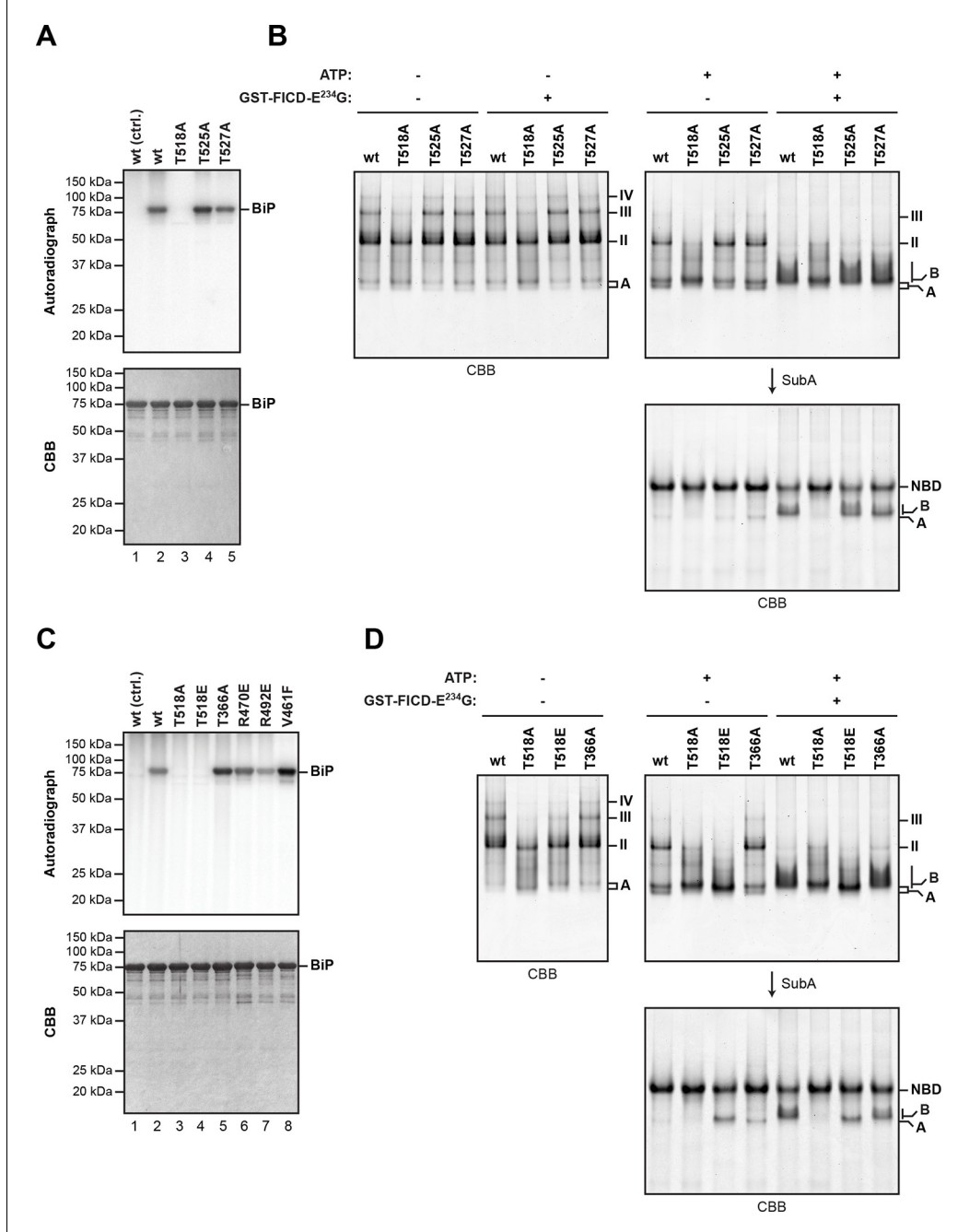

**Figure 5.** Mutation of threonine 518 in the substrate binding domain of BiP abolishes its AMPylation in vitro. (**A**) Autoradiograph and Coomassie (CBB) stain of an SDS-PAGE gel of recombinant bacterially-expressed wildtype (wt) BiP and the indicated mutants exposed in vitro to active GST-FICD[E234G] coupled to GSH-Sepharose beads (lanes 2-5) or GST alone as a control (lane 1) in the presence of $\alpha$-$^{32}$P-ATP as a substrate. (**B**) Coomassie-stained native gel of wildtype BiP and the indicated mutants (all at 20 µM), following exposure to ATP (1.5 mM), GST-FICD[E234G] (0.8 µM), both or neither (for 45 min at 30°C). Where indicated the samples were afterwards exposed to SubA (30 ng/µl, 10 min at room temperature). (**C**) As in "A" above, with a different set of mutant BiP proteins. (**D**) As in "B" above, with a different set of mutant BiP proteins. Note that both the T518E and T518A mutations (in panel "B" above) affect the mobility of the 'A' form of BiP and forestall further changes in mobility by FICD, but only the T518E mutation mimics enzyme-mediated AMPylation by promoting a 'B' form-like state partially resistant to cleavage by SubA.

conformation is the preferred substrate for FICD. The recently determined crystal structure of human BiP substrate binding domain supports this conclusion in that the loop encompassing the Thr[518] AMPylation site is stabilized by six polar interactions present in the apo/ADP conformation that are

lost in the ATP conformation (*Yang et al., 2015*), freeing the Thr$^{518}$ side-chain to react with the active site of FICD (*Figure 7—figure supplement 2*).

## Consequences of AMPylation to BiP function in vitro

Impaired oligomerization of AMPylated BiP and its restricted conformational flexibility (reflected in resistance to cleavage by SubA) suggest that the modification affects BiP function and activity. Hsp70s have low basal ATP hydrolysis rates that are stimulated by J protein co-factors (*Liberek et al., 1991*). Therefore, we compared the basal and J protein-stimulated ATPase activity of unmodified BiP and BiP modified to completion. AMPylation lowered the basal ATP hydrolysis rate of BiP by ~50% (*Figure 8A and B*) and, while the ATPase activity of unmodified BiP was enhanced by the presence of a J-domain in a concentration-dependent manner, AMPylated BiP was almost entirely resistant to J-mediated stimulation of ATP hydrolysis (*Figure 8C*).

AMPylation barely influenced steady-state binding of substrate peptide to BiP: The dissociation constants of the complex between unmodified BiP or AMPylated BiP and a well-characterized substrate peptide, HTFPAVLGSC, measured at equilibrium (in the presence of ADP) by fluorescence polarization, were similar ($K_d^{BiP}$ 16.7 μM and $K_d^{BiP-AMP}$ 11.5 μM; *Figure 8D*) and within the range previously reported (*Chambers et al., 2012*; *Marcinowski et al., 2011*). However, challenge with excess of unlabeled peptide revealed a markedly higher "off" rate of the complex between AMPylated BiP and the bound peptide ($k_{off}^{BiP}$ 0.037 ± 0.006 min$^{-1}$ vs. $k_{off}^{BiP-AMP}$ 0.212 ± 0.021 min$^{-1}$; *Figure 8E*).

These observations support the view that AMPylated BiP preferentially populated an ATP-like state with high substrate "on" and "off" rates, even when associated with ADP, whereas in the ATP-bound state AMPylated BiP was rendered inactive by its inability to respond to J protein co-factor.

## Enhanced buffering capacity of the FICD-deficient ER

AMPylation weakens the interaction of BiP with its substrates and was therefore hypothesized to reverse BiP's ability to repress the UPR transducers - whether imposed directly by binding to their regulatory lumenal domains (*Bertolotti et al., 2000*) or indirectly by competing for unfolded protein substrates (*Pincus et al., 2010*), BiP repression of the UPR involves stable engagement of clients in its substrate binding domain. To test the derivative prediction that enforced expression of an active FICD would lead to UPR activation, wildtype FICD or a catalytically-dead or active FICD$^{E234G}$ were introduced by transient transfection into CHO-K1 cells bearing an integrated *CHOP::GFP* UPR reporter. The dormant reporter was activated by tunicamycin-induced ER stress (*Figure 9A*, panels 1 and 2; a positive control), and by acquisition of a constitutively active FICD$^{E234G}$ but not by a catalytically dead FICD$^{E234G-H363A}$ nor by the regulated wildtype enzyme (*Figure 9A*, panels 3-5). The robustness of these observations and the independence of UPR activation from endogenous FICD – an important point given the evidence for FICD dimerization (*Bunney et al., 2014*) – are attested to by similar observations made in *FICD* knockout cells (*Figure 9—figure supplement 1*).

The induction of the UPR reporter in CHO-K1 cells was mirrored by the behavior of endogenous BiP, whose levels increased in a time-dependent manner in human HEK 293 cells upon activation of a conditional allele encoding FICD$^{E234G}$ (*Figure 9B*). Qualitatively similar observations were made in two separate clones of FICD$^{E234G}$-expressing cells [*Figure 9C*; the conspicuous basal (Dox-independent) SubA resistance of endogenous BiP, lanes 5 and 7, likely reflecting basal leakiness of the expression system]. These findings are readily explained by the engagement of a feedback mechanism (the UPR) that defends a level of active BiP and compensates for the accumulation of inert, AMPylated BiP ('B' form). As suggested previously (*Sanyal et al., 2015*), the capacity of the cells to compensate for the consequences of deregulated FICD activity is limited; reflected here by the adverse effect of sustained FICD$^{E234G}$ expression on cell viability (*Figure 9D*).

The aforementioned observations support the inactivating nature of FICD-mediated BiP modification and suggest that in some circumstances cells deficient in FICD may have elevated levels of active BiP. Given the strong repressive role of BiP on the UPR, enhanced availability of active chaperone attendant upon the absence of FICD and the consequential enhanced capacity to buffer unfolded protein stress might attenuate UPR signaling. Strong feedback control operative in the UPR specifies that such attenuation, were it to occur, would be transient and so more likely to be detected by monitoring early events in the UPR, such as PERK and IRE1α autophosphorylation, than

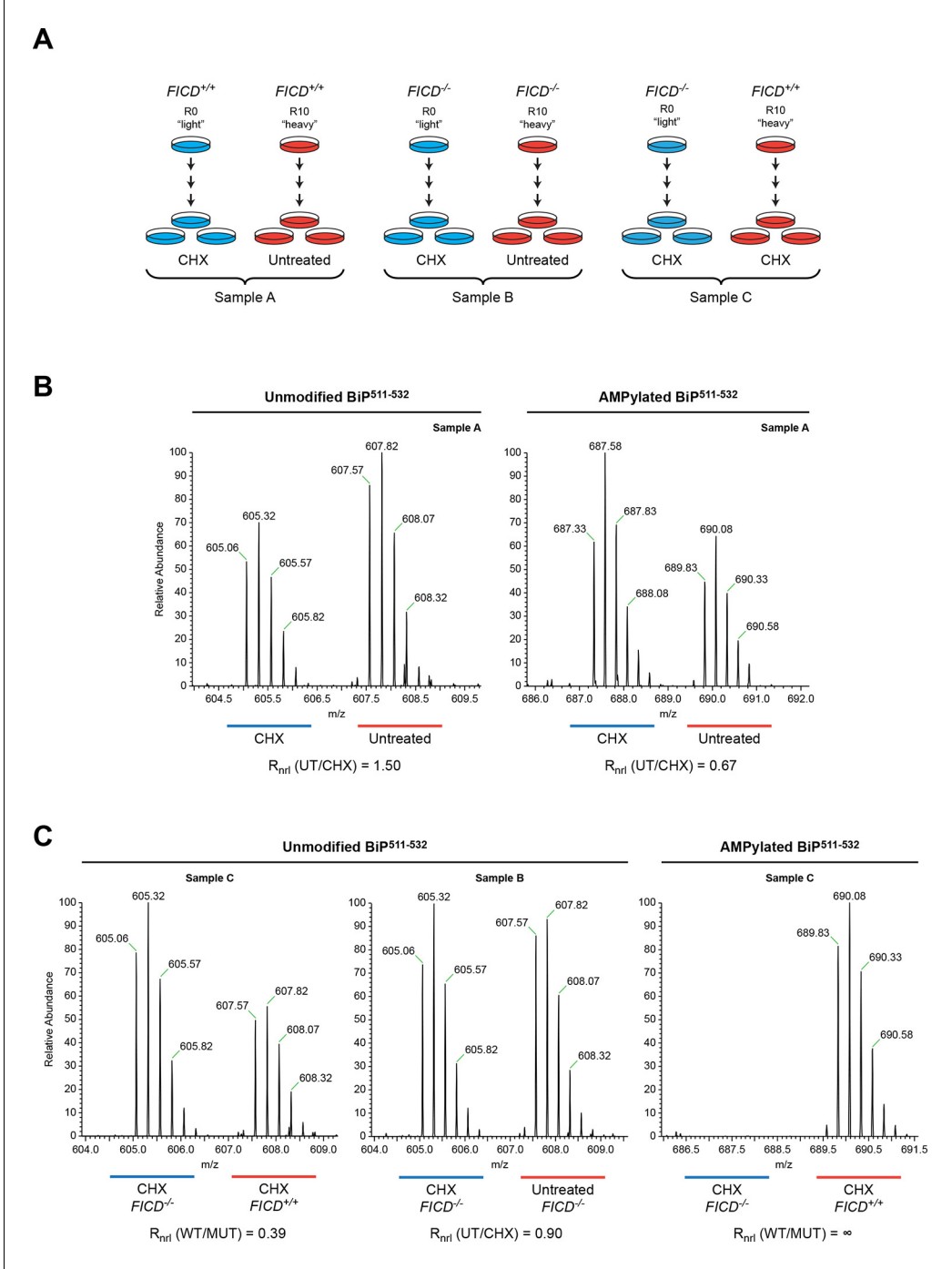

**Figure 6.** Reciprocal loss of unmodified and gain of AMPylated BiP[511-532] purified from CHO-K1 cells treated with cycloheximide. (**A**) Schema of the design of the SILAC experiment to quantify relative changes in abundance unmodified and AMPylated BiP peptides from untreated and cycloheximide (CHX)-treated wildtype and *FICD*[-/-] mutant CHO-K1 cells. (**B**) LC-MS spectra of unmodified and modified quadruply-charged BiP[511-532] peptides from a SILAC experiment where untreated "heavy" and cycloheximide-treated "light" samples from wildtype CHO-K1 cells were digested by Arg-C. The raw peptide abundance measurements were normalized to the recovery of a doubly-charged reference peptide, BiP[61-74], from the same SILAC samples (*Figure 6—figure supplement 1*) to arrive at the normalized ratio of the signal in the paired samples ($R_{nrl}$). Note that unmodified BiP[511-532] is depleted and AMPylated BiP[511-532] is reciprocally enriched in cycloheximide-treated wildtype cells. (**C**) Analysis as in "B" (above) applied to the indicated paired SILAC samples. Note that unmodified BiP[511-532] is depleted by cycloheximide treatment only in wildtype cells and AMPylated BiP[511-532] is only detected in wildtype cells. These observations were reproduced in a second independent SILAC experiment.

The following figure supplements are available for figure 6:

*Figure 6 continued on next page*

*Figure 6 continued*

**Figure supplement 1.** No change detected in abundance of unmodified BiP[337-367] purified from CHO-K1 cells treated with cycloheximide.

**Figure supplement 2.** Fragmentation spectra of unmodified and AMPylated BiP Arg-C peptide 511-532 pinpoints AMPylation to Thr[518].

later events, such as reporter gene activity. Furthermore, "over-chaperoning" attendant upon the absence of FICD would likely be revealed by manipulating the flux of proteins into the ER of a secretory cell. These theoretical considerations led us to AR42j pancreatic acinar cells - a secretory cell type in which PERK and IRE1α autophosphorylation are easy to detect (*Bertolotti et al., 2000*).

As expected, *FICD* inactivation by CRISPR-Cas9-mediated gene editing in AR42j cells eliminated both the acidic form of BiP from IEF-PAGE and the 'B' form from native-PAGE (*Figure 10A-C* and *Figure 10—figure supplement 1B*). Wildtype and *FICD*-lacking AR42j cells were exposed to cycloheximide to build up pools of modified BiP (in the wildtype cells) and then to the ER stress-inducing agent DTT, whose ability to activate the UPR does not require ongoing protein synthesis (*Bertolotti et al., 2000*). *FICD* knockout led to a conspicuous temporal delay in PERK and IRE1α autophosphorylation, observed in two independently derived AR42j *FICD^{-/-}* clones (*Figure 10D*). Sluggish activation of the UPR in the *FICD^{-/-}* cells was also reflected in attenuated PERK-dependent repression of protein synthesis imposed by DTT on cells pre-treated with cycloheximide. In this experiment, wildtype or *FICD^{-/-}* AR42j cells were pre-exposed to cycloheximide for 3 hr followed by brief washout to allow recovery of protein synthesis, which was measured by incorporation of puromycin label into newly synthesized proteins. In wildtype cells, DTT led to marked attenuation of protein synthesis, regardless of pre-exposure to cycloheximide (*Figure 10E*, compare lanes 1 and 2 and 3 and 4), however pre-exposure to cycloheximide attenuated subsequent translational repression by DTT-mediated PERK activation in *FICD^{-/-}* cells (*Figure 10E*, lanes 5-8).

The aforementioned defect in UPR activation in *FICD^{-/-}* cells was both fleeting and dependent on the pre-imposition of a low incoming protein flux regime on the ER (attained experimentally by cycloheximide pre-treatment), as *FICD* deletion had no measureable effect on the kinetics of *CHOP:: GFP* expression induced by treatment with tunicamycin (*Figure 10—figure supplement 2*). These observations are inconsistent with the role proposed for FICD in signaling by the PERK branch of the UPR (*Sanyal et al., 2015*) and suggest instead that FICD-mediated BiP inactivation adjusted the level of active BiP to transient fluctuations in unfolded protein flux into the ER. In the absence of this mechanism, *FICD* knockout cells experienced an excess of functional chaperone, which was revealed here as attenuated activation of the earliest steps of the UPR.

## Discussion

Previous glimpses at BiP modification have been largely indirect: Inferred from transfer of radiolabel from intracellular $^{32}$P phosphate or $^{3}$H adenosine pools to an acidic form of BiP and have led to the conclusion that the modification(s) consists of phosphorylation or ADP-ribosylation or both. Attempts to pinpoint these hypothesized modifications directly were unsuccessful. The discovery of FICD-mediated BiP AMPylation (*Ham et al., 2014*; *Sanyal et al., 2015*) was thus an important breakthrough whose significance is enhanced further by our finding that elimination of FICD abolishes all evidence for endogenous BiP modification in vivo as detected by IEF- and native-PAGE, and changes in mass of the endogenous protein. These findings unify and clarify the pre-existing observations regarding BiP modification and indicate that FICD-mediated AMPylation on Thr[518] is the major, if not the only covalent change in this chaperone detectable by the existing methods. Furthermore, FICD-mediated AMPylation on Thr[518] is sufficient to account for the previously observed key features of modified BiP, namely its acidity and inertness.

Our findings are consonant with metabolic labeling experiments that mapped the modification to the C-terminal substrate binding domain of BiP and provide positive evidence for absence of significant FICD-mediated modification of Ser[365] or Thr[366], neither in vitro nor in vivo. Provided ATP as a co-substrate, FICD modifies BiP in vitro at a single site with a mass expected of AMPylation. AMPylation maps solely to peptides encompassing Thr[518] and these are modified to high stoichiometry in vitro. Furthermore, an unbiased chemical proteomic approach to profiling proteins that are modified

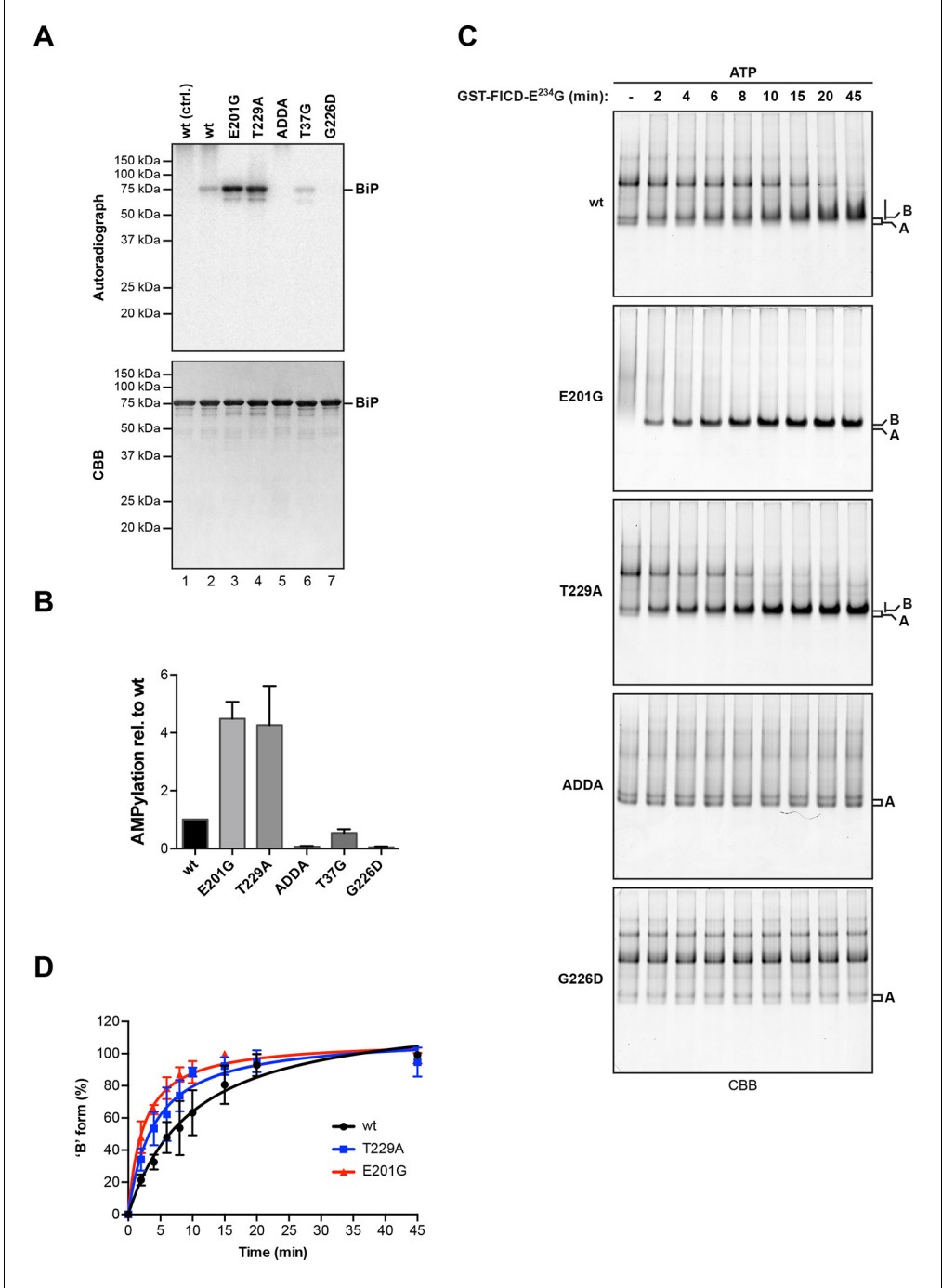

**Figure 7.** AMPylation of BiP is sensitive to its conformational state. (**A**) Autoradiograph and Coomassie (CBB) stain of an SDS-PAGE gel of wildtype (wt) BiP and the indicated mutants exposed in vitro to active GST-FICD$^{E234G}$ coupled to GSH-Sepharose beads (lanes 2-5) or GST alone as a control (lane 1) in the presence of $\alpha$-$^{32}$P-ATP as a substrate. (**B**) Bar graph of densitometric quantification of radiolabeled BiP proteins from in vitro AMPylation reactions as in "A". The radioactive signals were normalized to the amount of loaded protein (CBB signal) and the values for wildtype BiP protein were arbitrarily set to 1. Mean values ± SD of three independent experiments are shown. (**C**) Coomassie-stained native gels of the indicated BiP mutants (all at 20 µM) following exposure to bacterially expressed GST-FICD$^{E234G}$ (0.8 µM) in presence of 1.5 mM ATP for the indicated time. Note the reduced ability of BiP$^{E201G}$ to form discrete oligomers in presence of ATP, which may be due to altered substrate interaction characteristics. Also note the sharpness of the bands of modified BiP$^{E201G}$ and BiP$^{T229A}$, which suggest a high degree conformational uniformity and strongly reduced substrate interactions. (**D**) Plot of time-dependent accumulation of the 'B' form of BiP from experiments as shown in "C". Initial values were set to 0% and end-point values were set to 100% for each of the BiP versions, respectively. Mean values ± SD of three independent experiments are shown. Non-linear

*Figure 7 continued on next page*

*Figure 7 continued*

regression analysis was performed to determine $t_{1/2max}$ values, which were 9.9 min, 4.0 min and 2.3 min for wildtype BiP, BiP$^{T229A}$ and BiP$^{E201G}$, respectively.

The following figure supplements are available for figure 7:

**Figure supplement 1.** The isolated BiP substrate binding domain is not measurably AMPylated by FICD in vitro.

**Figure supplement 2.** Loop 7,8 of the BiP substrate binding domain is destabilized in the ATP-bound conformation.

by FICD in a cell lysate in vitro led to the identification of BiP Thr$^{518}$ as an AMPylation site, whereas no modification at Ser$^{365}$ or Thr$^{366}$ was reported (*Broncel et al., 2015*). It is thus tempting to speculate that AMPylation on Thr$^{518}$ is the only quantitatively significant modification of BiP carried out by FICD in vivo too. However, the discovery that an FIC-domain containing bacterial enzyme AMPylates eukaryotic targets (*Kinch et al., 2009*; *Yarbrough et al., 2009*) was closely followed by the realization that the domain may also participate in transfer of part of other pyrophosphate-containing metabolites, leading to UMPylation, mono-phosphorylation and conjugation of phosphocholine (reviewed in *Garcia-Pino et al., 2014*). Thus, one must keep an open mind in regards to the possibility of parallel FICD-mediated modification of BiP (and possibly other ER proteins) by metabolites (other than ATP) found in the ER.

The phenotype of *FICD* deletion in *Drosophila* is restricted to a defect in light sensing and involves disruption of neurotransmitter (histamine) re-uptake via glial cells. The subcellular loci of *Drosophila* FICD action are unclear: Consistent with an ER-based function, the protein co-fractionates with BiP. However, HRP-tagged endogenous FICD led to prominent staining of capitate projections, a glial cell plasma membrane component of photoreceptor synapses (*Rahman et al., 2012*). Therefore, it is possible that FICD may have substrates other than BiP that figure prominently in its action in specific cell types.

Thr$^{518}$ of hamster BiP is highly conserved in other eukaryotes. The crystal structure of the isolated substrate binding domain of human BiP (PDB 5E86) and yeast Kar2/BiP (PDB 3H0X) (locked in the ADP-like state), shows the corresponding residues – human Thr$^{518}$ and Kar2 Thr$^{538}$ – to be located on a loop connecting beta strands 7 and 8 (L$_{7,8}$), with its side-chain stabilized by a network of polar contacts with the conserved side chain of Asp$^{515}$/Asp$^{535}$ and the backbone amine of Asn$^{520}$/Lys$^{540}$. However, both interactions are disrupted in the substrate binding domain of human BiP in the ATP state (PDB 5E84). Exposure of the Thr$^{518}$ side chain to the solvent is likely enhanced further by the loss of four other polar interactions that stabilize L$_{7,8}$ in the ADP state (*Yang et al., 2015*). Therefore, mobilization of the side chain by allosteric signals attendant upon ATP binding is suggested to enhance exposure of Thr$^{518}$ to solvent explaining the selective FICD-mediated modification of ATP-bound BiP and the absence of modification of mutant variants of BiP that are defective in domain coupling (BiP$^{ADDA}$, BiP$^{G226D}$ and to a lesser extent BiP$^{T37G}$). These recent observations on human BiP (*Yang et al., 2015*) also showcase the long range allostery common to all Hsp70 proteins (*Mayer, 2013*): the identity of the nucleotide bound in the nucleotide binding domain affects stability of L$_{7,8}$ on the opposite side of the protein. It seems plausible that a modification affecting the disposition of this loop could reach the nucleotide binding domain through a reciprocal set of allosteric interactions to explain the effects of AMPylation on nucleotide hydrolysis and responsiveness to J proteins.

An earlier study from our lab revealed that mutations affecting residues R470 and R492 consistently diminished labeling of overexpressed mutant BiP from $^{32}$P orthophosphate or $^{3}$H adenosine pools in vivo (*Chambers et al., 2012*), leading us to suggest that these residues undergo ADP-ribosylation in vivo. The current findings indicate that our earlier conclusions were in error and suggest that the important structural role of R470 [in all Hsp70s (*Fernandez-Saiz et al., 2006*)] and R492 [specifically in BiP (*Yang et al., 2015*)] may have given rise to defective allosteric transitions by the R470 and R492 BiP mutants. Such defects appear to have a more conspicuous effect on modification of overexpressed BiP in vivo than on the modification of pure recombinant BiP by FICD in vitro.

The structures of nucleotide-bound Hsp70s also provide hints to the functional consequences of BiP AMPylation. Residues comprising beta strand 8 in the substrate binding domain of the ADP-

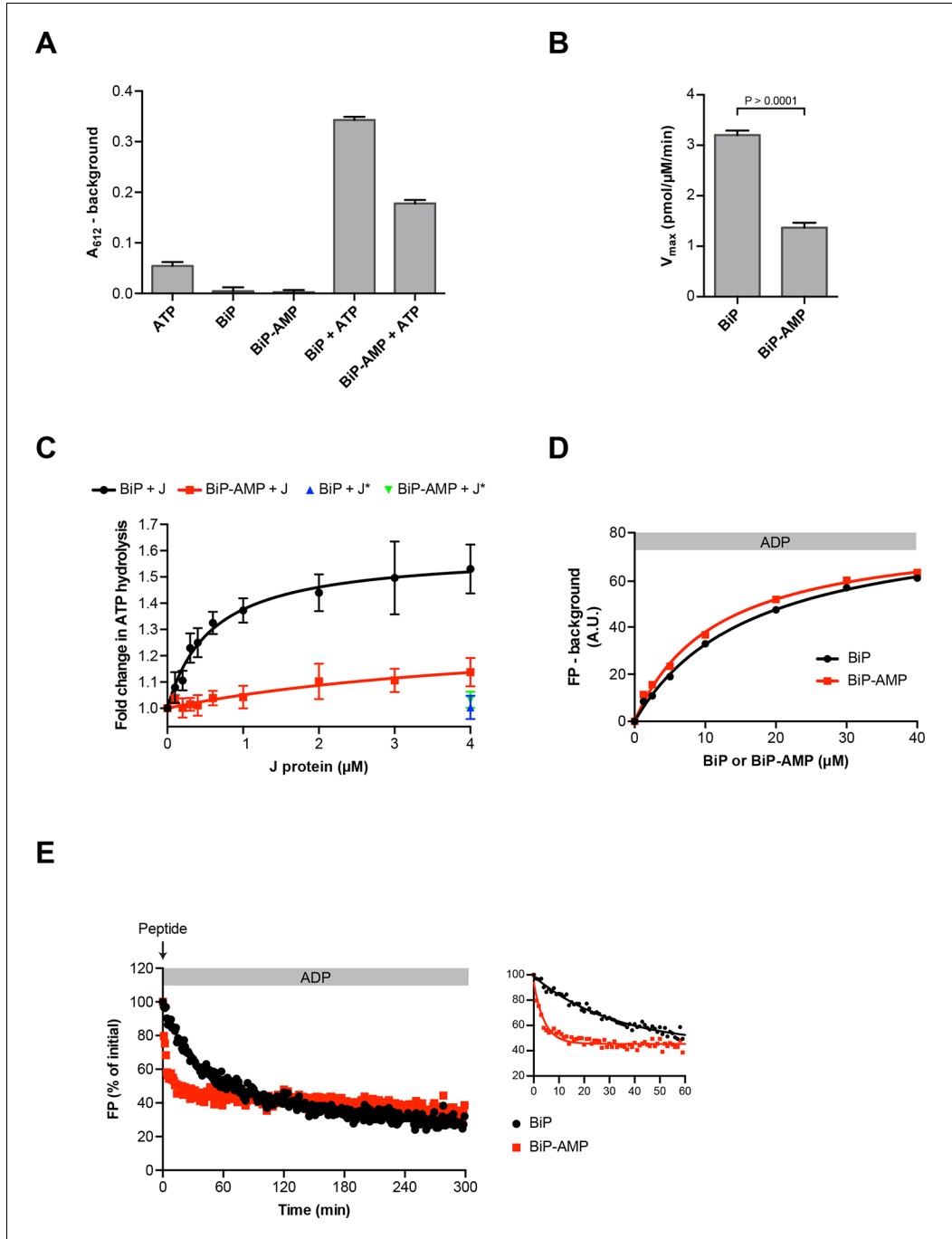

**Figure 8.** Functional consequences of BiP AMPylation in vitro. (**A**) Bar diagram of ATP hydrolysis by BiP and BiP AMPylated to completion (BiP-AMP), as reflected in phosphate release (detected colorimetrically). Samples containing either purified BiP or BiP-AMP (both at 5 µM) were incubated with 3 mM ATP for 1 hr at 30°C and free orthophosphate generated by ATP hydrolysis was measured. Samples lacking BiP or ATP report on the assay background. Bar graph shows mean absorbance values ± SD at 612 nm ($A_{612}$) of the complex between free orthophosphate and the malachite green dye after background subtraction of three repeats (n = 3). (**B**) Bar diagram of $V_{max}$ values for basal ATPase activities of unmodified and AMPylated BiP, derived from experiments as in "A". Mean values ± SD are shown (n = 3). (**C**) Measurement of J protein-stimulated ATPase activity of unmodified and AMPylated BiP. Samples of purified BiP or BiP-AMP (both at 1.5 µM) were incubated in presence of 2 mM ATP with isolated J-domain of ERdj6 (J) at the indicated concentrations for 3 hr at 30°C and released orthophosphate was detected as in "A". The control reactions contained 4 µM of non-functional ERdj6$^{H422Q}$ J-domain (carrying a mutation in the critical HPD motif; J*) instead of the wildtype J-domain. Shown is the J protein-dependent change in ATP hydrolysis rate of BiP and BiP-AMP relative to their basal ATP hydrolysis rates in absence of J protein (set to 1) of four experiments (values ± SD, n = 4). (**D**) Plot of concentration-dependent steady-state binding of substrate peptide by unmodified or AMPylated BiP. Fluorescence polarization (FP) of 1 µM lucifer yellow-labeled BiP substrate peptide (HTFPAVLGSC) was measured after incubation with purified BiP or BiP-AMP at the

*Figure 8 continued on next page*

Figure 8 continued

indicated concentrations for 24 hr at 30°C in presence of 1 mM ADP. Mean values of a representative experiment performed in triplicates are shown. (E) Plot of time-dependent release of fluorescently-labeled substrate peptide from unmodified or AMPylated BiP, following injection of 400-fold excess of unlabeled substrate peptide. Fluorescence polarization (FP) signal of lucifer yellow-labeled substrate peptide (1 µM) bound to BiP or BiP-AMP (both at 40 µM) in presence of 1 mM ADP (as in "D") was measured after addition of 400-fold excess of unlabeled substrate peptide (0.4 mM) at t = 0. The initial values (after background subtraction) were set to 100% and non-linear regression analysis was performed on the first 60 min of peptide competition (inset). Mean values of a representative experiment performed in triplicates are plotted on the graph. The mean dissociation rate constants ($k_{off}$) ± SD for BiP = 0.037 ± 0.006 $min^{-1}$ and BiP-AMP = 0.212 ± 0.021 $min^{-1}$ as well as the mean half-lives ($t_{1/2}$) ± SD for BiP = 19.3 ± 2.9 min and BiP-AMP = 3.3 ± 0.3 min were calculated based on three independent experiments.

The following source data is available for figure 8:

**Source data 1.** Data from three independent repeats (each performed in triplicates) of the experiment presented in *Figure 8E* are shown.

bound form of BiP and DnaK are dramatically delocalized in the ATP-bound form (*Kityk et al., 2012*; *Qi et al., 2013*; *Yang et al., 2015*) enabling the domain movements that underlie the functional allosteric transitions (*Mayer, 2013*). By altering the conformation of the preceding loop, AMPylation of Thr[518] could facilitate the melting of beta strand 8, thus favoring a new state of the chaperone. Our observations suggest that this new state would resemble the ATP-bound state in terms of high substrate "off" rates, but would differ from it in terms of (un)responsiveness to J protein-driven ATP hydrolysis. It is notable in this regard that the ATP-like conformation can be uncoupled from responsiveness to J protein, as a mutation of BiP's substrate binding domain (G461P/G468P) that locks the protein in an ATP-like conformation is also refractory to J protein-mediated stimulation of ATP hydrolysis (*Yang et al., 2015*).

These structural considerations fit with the established inverse correlation between activity of BiP (imposed by the burden of client proteins) and the extent of modification: Nutrient deprivation and protein synthesis inhibitors, which lower the flux of unfolded proteins into the ER, increase the acidic form of BiP in cultured cells (*Laitusis et al., 1999*; *Ledford and Jacobs, 1986*) and animal tissues (*Chambers et al., 2012*), whereas imposition of ER stress leads to less modified BiP (*Chambers et al., 2012*; *Ham et al., 2014*; *Hendershot et al., 1988*; *Laitusis et al., 1999*; *Leno and Ledford, 1989* and our observations here). All this suggests a simple mechanism whereby substrate-free, ATP-associated BiP partitions between two mutually exclusive fates: entering the substrate binding cycle or AMPylation. The former is driven by the concentration of unfolded client proteins and catalyzed by J proteins, whereas the latter imposes on BiP an inactive conformation that disfavors the J protein-mediated ATP hydrolysis required for high affinity client binding (*De Los Rios and Barducci, 2014*; *Misselwitz et al., 1998*). High client "off" rates ensure that AMPylated BiP would neither interfere with protein folding by interacting excessively with substrates, nor would it unduly repress the UPR transducers. And futile cycles of ATP hydrolysis would be obviated by refractoriness to stimulation by J protein. However, given the reversibility of the modification, this inert BiP would serve as a repository for active chaperone were it needed (*Figure 11*, a model).

Whilst such a simple kinetic competition model could account for the inverse relationship between unfolded protein load and AMPylation, there are hints of other layers of refining regulation. Like other type II or III FIC domain-containing proteins, FICD is intrinsically repressed by the insertion of a conserved glutamate into the ATP binding pocket, delocalizing the co-substrate ATP (*Bunney et al., 2014*; *Engel et al., 2012*). Relief of such repression, imposed experimentally by the E234G mutation (*Engel et al., 2012*) or by a nanobody (*Truttmann et al., 2015*), is likely attained physiologically through allosteric mechanisms. Evidence that FICD binds unfolded proteins directly (*Sanyal et al., 2015*) hint at the possibility that such allostery might be responsive to the burden of unfolded proteins in the ER or might be regulated by FICD's oligomeric state or co-factor binding (*Bunney et al., 2014*). Coupling of FICD's intrinsic activity to protein folding homeostasis in the ER could conserve energy by limiting the futile cycles of AMPylation and de-AMPylation prescribed by regulation based solely on kinetic competition with substrates for BiP in the ATP-bound state. Furthermore, regulation of FICD enzymatic activity would explain the dissociation between transcriptional induction of the *FICD* gene during the UPR and the emergence of modified BiP, which, as shown here, is delayed until after resolution of the stress.

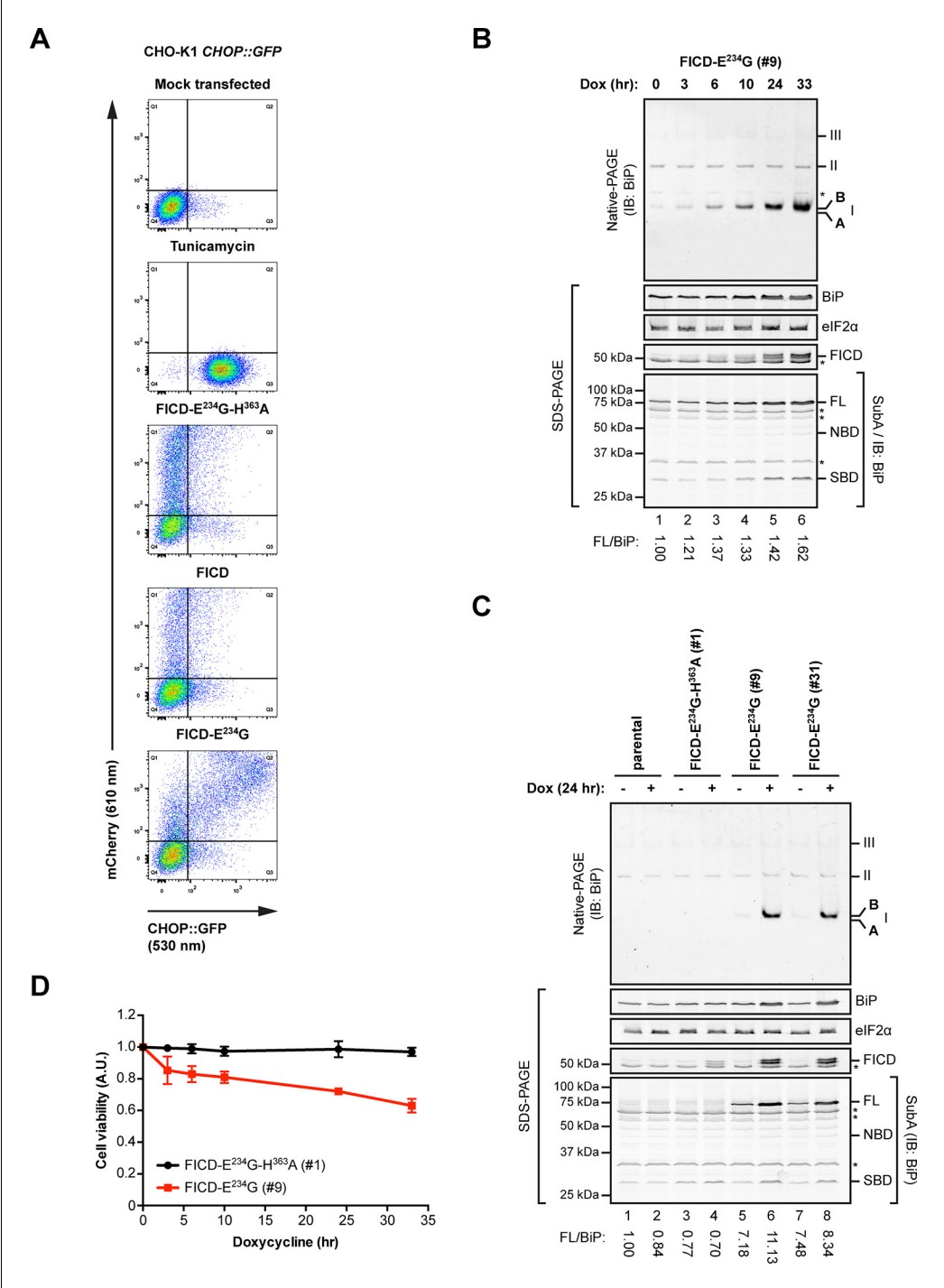

**Figure 9.** Overexpression of active FICD$^{E234G}$ activates the UPR. (**A**) Flow cytometry analysis of CHO-K1 *CHOP::GFP* UPR reporter cells transiently transfected with plasmids encoding wildtype FICD, the constitutively active FICD$^{E234G}$ or the inactive FICD$^{E234G-H363A}$ mutant alongside an mCherry transfection marker. Mock transfected cells that were treated with the UPR-inducing compound tunicamycin (2.5 μg/ml) for 16 hr before the analysis were included as controls. Note the accumulation of CHOP::GFP-positive (and mCherry-negative) cells in Q3 in the untransfected tunicamycin-treated samples and enhanced UPR reporter activation in cells transfected with plasmid encoding active FICD$^{E234G}$ and a co-expressed mCherry marker (reflected in the large number of double-positive cells in Q2). (**B**) Native gel immunoblot of endogenous BiP from lysates (supplemented with 1 mM ATP) of Flp-In T-REx 293 cells that carry a stable transgene encoding a doxycycline-inducible form of the active FICD$^{E234G}$ mutant. The cells were treated for the indicated time with doxycycline (Dox) prior to lysis. The major species visible on the native gel are numbered by order of descending mobility (I–III) and the monomeric 'B' and 'A' forms are marked. Immunoblots of the same samples resolved by SDS-PAGE report on FICD$^{E234G}$ expression, total BiP loaded and on eIF2α as a loading control. In addition, samples of the lysates were treated with SubA (30 ng/μl) for 10 min at room
*Figure 9 continued on next page*

*Figure 9 continued*

temperature before separation of proteins by SDS-PAGE and immunoblotting. Full-length BiP (FL), the nucleotide binding domain (NBD) and the substrate binding domain (SBD) are indicated. The ratios between the quantified signals of full-length BiP and total BiP (following cleavage by SubA) were normalized to the value observed in lane 1 (arbitrarily to 1) and are indicated below. The asterisks mark bands of unknown identity. Note the correlation between FICD[E234G] expression, the appearance of the monomeric 'B' form of BiP on the native gel as well as the increasing resistance of BiP towards cleavage by SubA. (C) Analysis of Flp-In T-REx 293 cells upon doxycycline-induced expression of inactive FICD[E234G-H363A] or active FICD[E234G] as in "B" above. (D) Proliferation assay with Flp-In T-REx 293 cells upon doxycycline-induced expression of inactive FICD[E234G-H363A] or active FICD[E234G] for the indicated times. Shown are mean values ± SD relative to uninduced cells (set to 1) of three independent experiments ($n = 3$).

The following figure supplement is available for figure 9:

**Figure supplement 1.** Overexpression of active FICD[E234G] induces UPR in *FICD[-/-]* cells.

Recent findings suggest the existence of an alternative inactive state of BiP effected by oligomerization. The architecture of the oligomers, whereby one BiP molecule engages the interdomain linker of another as a conventional substrate peptide (*Preissler et al., 2015*), implies that BiP protomers are locked in the ADP-bound state and unlikely to serve as a substrate for AMPylation. This conjecture fits the observation that depletion of ER calcium, which promotes BiP oligomerization at the expense of substrate binding, is associated with gradual depletion of the FICD-dependent modified 'B' form of BiP (*Preissler et al., 2015*).

The benefits of ensuring an adequate reserve of chaperones to cope with the unfolded protein burden are clearly revealed by the consequences of defects in UPR signaling (*Walter and Ron, 2011*). But the emergence of active mechanisms to downregulate BiP activity suggest that an excess of functional BiP might have a cost too. This cost is less obvious than that associated with UPR defects, as FICD inactivation results in no conspicuous growth defect in cultured cells. Nonetheless, our findings indicate that in circumstances of limited client protein load, the absence of FICD shifts the equilibrium in the ER in favor of chaperones over their clients (at least as reflected in attenuated UPR signaling). BiP overexpression works in the same direction and is known to impose a measure of inefficiency in the secretion of certain proteins (*Dorner et al., 1992*), it will thus be interesting to learn if FICD inactivation also promotes inefficiency in secretion and if so, under what circumstances.

## Materials and methods

### Plasmid construction

*Supplementary file 1* lists the plasmids used, their lab names, description and notes their first appearance in the figures and their corresponding label, and provides a published reference, where available.

A combination of PCR-based manipulations, restriction digests and site-directed mutagenesis procedures were used to mobilize the coding sequences and produce in-frame fusions with the affinity tags [GST, hexahistidine (His6) and His6-Smt3 epitopes] or mCherry fluorescent marker, and to create the point mutations indicated in the text.

### Mammalian cell culture

All cells were grown in tissue culture dishes or multi-well plates (Corning) at 37°C and 5% $CO_2$ and the following cell line-specific media were used:

CHO-K1 cells (ATCC CCL-61) were phenotypically validated as proline auxotrophs and their *Cricitulus griseus* origin was confirmed by genomic sequencing. The cells were cultured in Nutrient mixture F-12 Ham (Sigma, UK) supplemented with 10% (v/v) serum (FetalClone II; HyClone, South Logan, UT), 1 x Penicillin-Streptomycin (Sigma) and 2 mM L-glutamine (Sigma).

AR42j cells (ATCC CRL-1492) were phenotypically validated by documenting inducibility of amylase expression by dexamethasone and their *Rattus* sp. origin was confirmed by genomic sequencing. They were cultured in DMEM (Sigma) supplemented with 10% (v/v) serum (FetalClone II; HyClone), 1 x Penicillin-Streptomycin (Sigma), 2 mM L-glutamine (Sigma), 1 x non-essential amino acids (Sigma) and 50 μM β-mercaptoethanol (Gibco; Life Technologies, UK).

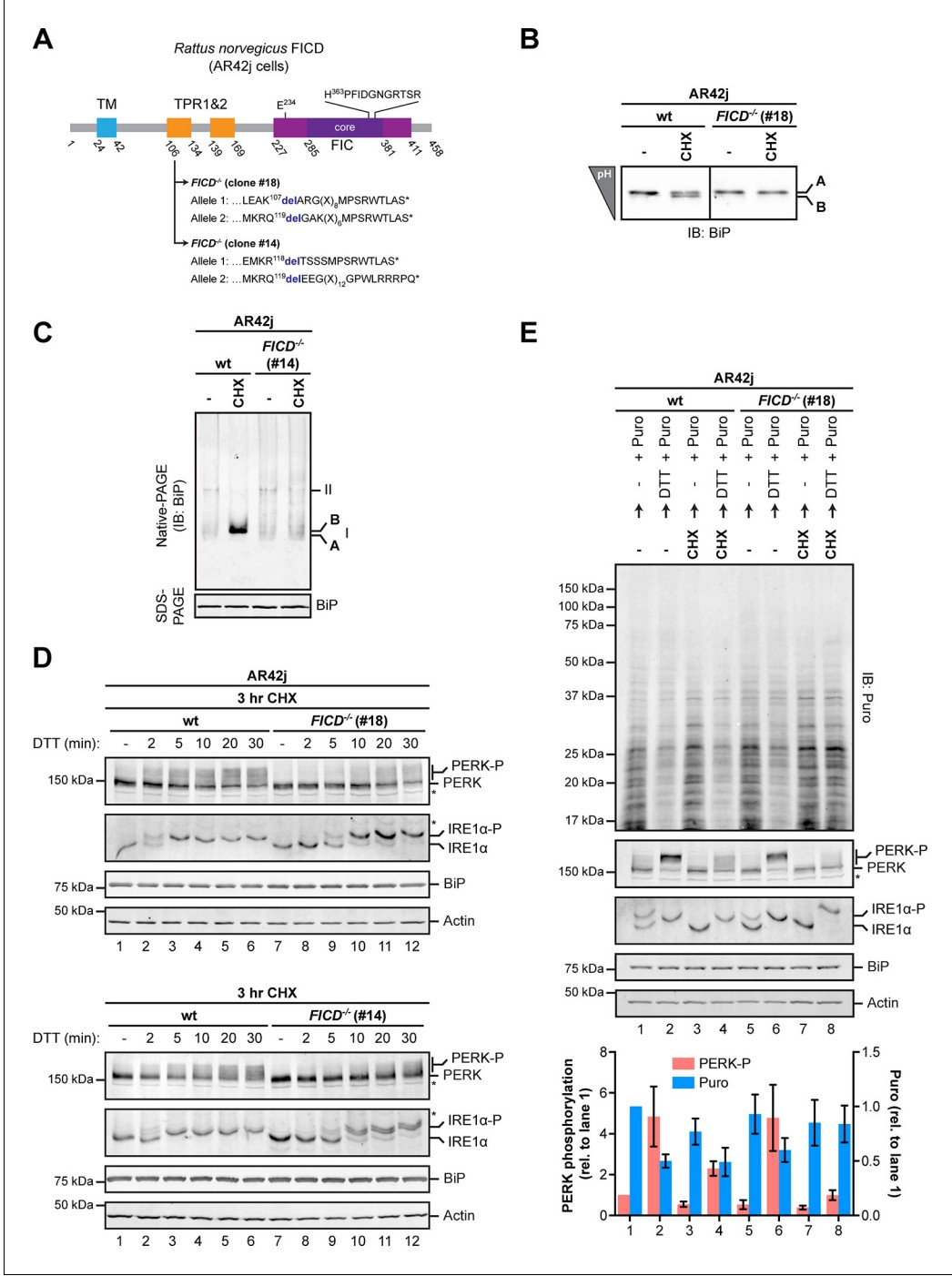

**Figure 10.** Over-chaperoning in *FICD*-deficient cells delays UPR signaling. (**A**) Schematic illustration of the rat FICD protein. Protein domains are highlighted and the mutations introduced by CRISPR-Cas9-mediated genome editing are presented as in *Figure 2A*. (**B**) Isoelectric focusing (IEF) followed by immunoblot of endogenous BiP from wildtype (wt) and *FICD*$^{-/-}$ AR42j cell lysates. (**C**) Native gel immunoblots of endogenous BiP from ATP-depleted wildtype and *FICD*$^{-/-}$ AR42j cell lysates (**D**) Endogenous PERK, IRE1$\alpha$, BiP and actin immunoblots of an SDS-PAGE gel on which lysates of wildtype and *FICD*$^{-/-}$ AR42j cells were resolved. The cells were pre-exposed to cycloheximide (CHX, 100 µg/ml) for 3 hr followed by treatment with 1 mM DTT for the indicated times. Slower migrating phosphorylated PERK is marked (PERK-P). To detect phosphorylated IRE1$\alpha$ (IRE1$\alpha$-P) samples of the same lysates were resolved on a Phos-tag gel. Actin served as a loading control. Asterisks mark bands of unknown identity. Note the delayed phosphorylation (activation) of PERK and IRE1$\alpha$ in the two *FICD*$^{-/-}$ clones. The experiment was performed four times with comparable results. (**E**) Immunoblots of puromycinylated proteins, PERK, IRE1$\alpha$, BiP and actin from wildtype and *FICD*$^{-/-}$ AR42j cells. Where indicated, the cells were pre-treated with CHX (100 µg/ml) for 3 hr followed by washout and exposure to puromycin (Puro, 10 µg/ml) in presence or absence of the reducing agent DTT (1 mM) for 30 min. Puromycin incorporation into nascent chains (reporting on protein synthesis rates), and PERK-P signals expressed relative to

*Figure 10 continued*

lane 1 (arbitrarily set to 1) are plotted in the bar graph below (mean values ± SD of three independent experiments, *n* = 3). The asterisk indicates a band of unknown identity. Note persistent protein synthesis in *FICD*⁻/⁻ cells that were exposed to DTT after CHX pre-treatment (lane 8). Also note the lower basal levels of phosphorylated forms of IRE1α and PERK in untreated *FICD*⁻/⁻ cells.

The following source data and figure supplements are available for figure 10:

**Source data 1.** Data from three independent repeats used for quantification shown in the graph in *Figure 10E*.

**Source data 2.** Source file of the flow cytometry data used to generate the plot in *Figure 10—figure supplement 2D*.

**Figure supplement 1.** Undetectable FICD protein in AR42j *FICD*⁻/⁻ cell lysates.

**Figure supplement 2.** Absence of FICD does not measurably affect the induction kinetics of the transcriptional response to unfolded protein stress.

HEK293T cells (ATCC CRL-3216) were cultured in DMEM (Sigma) supplemented with 10% (v/v) serum (FetalClone II; HyClone), 1 x Penicillin-Streptomycin (Sigma) and 2 mM L-glutamine (Sigma). Flp-In T-REx 293 cells (Invitrogen, UK) were cultured in the same medium except that 10% (v/v) tetracycline-free serum (FBS Premium; PAN-Biotech, Germany) was used.

All experiments with untransfected cells were performed at cell densities of 60-90% confluence. For pharmacological treatments the drugs were first diluted into pre-warmed culture medium, mixed and immediately applied to the cells by medium exchange. Unless indicated otherwise the following final concentrations were used: 100 μg/ml cycloheximide (Sigma), 0.5 μM thapsigargin (Calbiochem, Germany), 1 mM DTT, 2.5 μg/ml tunicamycin (Melford, UK), 10 μg/ml puromycin (Calbiochem), and 3 mM 2-deoxy-D-glucose (ACROS Organics, Belgium).

## *FICD* knockout using the CRISPR-Cas9 system

Three single guide RNA sequences (plasmids UK1448, UK1449 and UK1450; *Supplementary file 1*) for targeting the third exon of *Cricetulus griseus* (Chinese hamster) *FICD* were selected from the CRISPy database [URL: http://staff.biosustain.dtu.dk/laeb/crispy/, (*Ronda et al., 2014*)] and duplex DNA oligonucleotides of the sequences were inserted into the pSpCas9(BB)-2A-GFP plasmid (plasmid UK1359; *Supplementary file 1*) following published procedures (*Ran et al., 2013*). 2 x 10⁵ CHO-K1 or CHO-K1 *CHOP::GFP* reporter cells (*Novoa et al., 2001*) were plated in 6-well plates. Twenty-four hr later the cells were transfected with a combination of guide RNA/Cas9 plasmids UK1448 and UK1449 or UK1448 and UK1450 (2 μg total plasmid DNA per transfection) using Lipofectamine LTX (Invitrogen). Thirty-six hr after transfection the cells were washed with PBS, resuspended in PBS containing 4 mM EDTA and 0.5% (w/v) BSA, and GFP-positive cells were individually sorted by fluorescence-activated cell sorting (FACS) into 96-well plates using a MoFlo Cell Sorter (Beckman Coulter). Clones were then analyzed by a PCR-based assay to detect *FICD* mutations as described (*Klampfl et al., 2013*). Briefly, primers were designed for the region encompassing the *FICD* RNA guide target sites and the reverse primer was labeled with 6-carboxyfluorescein (6-FAM) on the 5' end. A PCR reaction was set up using 5 μl of AmpliTaq Gold 360 Master Mix (Applied Biosystems, UK), 0.6 μl of a mix of 10 μM forward and labeled reverse primers, 3.4 μl H₂O and 1 μl genomic DNA (approximately 10 ng/μl). PCR was performed as follows: 95°C for 10 min, 10 x (94°C for 15 s, 59°C for 15 s, 72°C for 30 s), 20 x (89°C for 15 s, 59°C for 15 s, 72°C for 30 s), 72°C for 20 min. PCR products were diluted 1:100 in water and fragment length was determined on a 3130xl Genetic Analyzer (Applied Biosystems) and the data were analyzed using the Gene Mapper software (Applied Biosystems). Clones for which frameshift-causing insertions or deletions were detected for both alleles were sequenced to confirm the *FICD* mutations.

*FICD* knockouts in the AR42j cell line were created as described above, using two single guide RNA sequences (plasmids UK1503 and UK1504; *Supplementary file 1*) for targeting the third exon of *Rattus norvegicus* (rat) *FICD* selected using the CRISPR Design tool [URL: http://crispr.mit.edu/ (Zhang Lab)]. These cells were electroporated using the Neon transfection system (Life Technologies, UK) as described (*Tsunoda et al., 2014*).

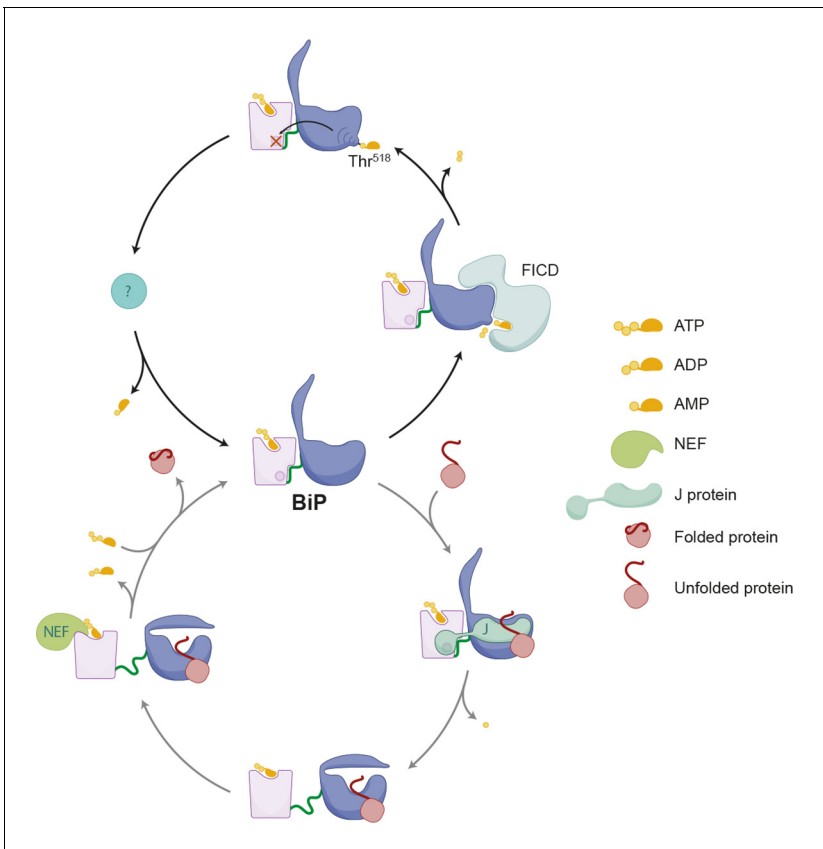

**Figure 11.** Schema depicting the hypothesized relationship between AMPylation and the BiP chaperone cycle. FICD-mediated AMPylation on Thr[518] allosterically traps BiP in a low substrate-affinity ATP-like state that is refractory to J protein-mediated stimulation of its ATPase activity. Removal of the modification by a phosphodiesterase allows BiP to re-join the chaperone cycle (depicted in the lower portion of the cartoon).

## Construction of stable cell lines expressing mutant versions of FICD

The effect of FICD overexpression was analyzed in mammalian cell clones carrying stable transgenes that encode doxycycline-inducible mutant versions of FICD. To generate stable cell lines, plasmid DNA encoding FICD[E234G] (UK1440) and FICD[E234G-H363A] (UK1446) were introduced into the Flp-In T-REx 293 cell line (Invitrogen) as described by the manufacturer's protocol. Briefly, *Hin*dIII-*Xho*I DNA fragments containing the coding sequences for FICD[E234G] and FICD[E234G-H363A] were sub-cloned into pcDNA5/FRT/TO (Invitrogen). The resulting expression plasmids were individually co-transfected along with the Flp-recombinase expression vector pOG44 (Invitrogen) using Lipofectamine 2000 (Invitrogen) into Flp-In T-REx 293 cells. Isogenic clones expressing the transgene under a doxycycline-inducible promoter were selected for resistance to blasticidin (3 µg/ml; Thermo Fisher, UK) and hygromycin (250 µg/ml; Invitrogen) and sensitivity to zeocin (50 µg/ml; Invitrogen).

## Cell proliferation assay

Flp-In T-REx 293 cells with stable transgenes encoding FICD[E234G] or FICD[E234G-H363A] were plated at a density of $8 \times 10^3$ cells per well on a 24-well tissue culture plate and grown for 24 hr. The cells were then treated with 0.1 µg/ml doxycycline (Melford) for the indicated times. Afterwards, the cells were washed once in regular medium and then maintained in regular medium for three days. Following the recovery period, the medium was replaced with fresh medium containing 0.02 mM WST-1 (Dojindo, Germany) and 0.02 mM 1-methoxy phenazine methosulfate (Sigma), and the cells were incubated for 60 min at 37°C before absorbance was measured at 440 nm. Each experiment was performed in duplicates and repeated three times.

## Flow cytometry

The effect of FICD overexpression (wildtype, FICD[E234G] and FICD[E234G-H363A] mutant versions) on the unfolded protein response was studied by transient transfection of wildtype and *FICD[-/-]* CHO-K1 *CHOP::GFP* UPR reporter cell lines with plasmids UK1397, UK1398 or UK1443 (see *Supplementary file 1*) using Lipofectamine LTX. Where indicated, cells were treated 24 hr after transfection with the ER stress-inducing agent tunicamycin (2.5 µg/µl) for 16 hr to activate *CHOP:: GFP* (which results in enhanced GFP production). Cells were analyzed by dual-channel flow cytometry with an LSRFortessa cell analyzer (BD Biosciences) as described previously (*Tsunoda et al., 2014*). GFP (excitation laser 488 nm, filter 530/30) and mCherry signals (excitation laser 561, filter 610/20) were detected. The data were processed using FlowJo and median reporter values were plotted using GraphPad Prism (GraphPad Software).

## Mammalian cell lysates

Cell lysis was performed as described (*Preissler et al., 2015*). Mammalian cells were grown on 10 cm dishes and treated as indicated. At the end of each experiment the dishes were placed on ice and washed twice with ice-cold PBS. The cells were detached with a cell scraper in PBS containing 1 mM EDTA, transferred to a 1.5 ml reaction tube and centrifuged for 5 min at 370 *g* at 4°C. The cells were lysed in HG lysis buffer [20 mM HEPES-KOH pH 7.4, 150 NaCl, 2 mM MgCl$_2$, 10 mM D-glucose, 10% (v/v) glycerol, 1% (v/v) Triton X-100] supplemented with protease inhibitiors [1 mM phenylmethylsulfonyl fluoride (PMSF), 2 µg/ml pepstatin, 2 µg/ml leupeptin, 4 µg/ml aprotinin] for 10 min on ice and centrifuged at 21,000 *g* for 10 min at 4°C. The protein concentrations of the cleared lysates were determined using the Bio-Rad protein assay reagent (Bio-Rad, Germany) and normalized with lysis buffer (usually the protein concentration was adjusted to 1.2 µg/µl). Where indicated the lysates were treated with SubA protease as described. For SDS polyacrylamide gel electrophoresis (SDS-PAGE) analysis, the proteins were denatured by addition of SDS sample buffer and heating for 10 min at 75°C followed by separation on 12% SDS polyacrylamide gels.

For detection of endogenous BiP by native-PAGE (see below) the cells were lysed in HG lysis buffer containing 2 x protease inhibitors (2 mM PMSF, 4 µg/ml pepstatin, 4 µg/ml leupeptin, 8 µg/ml aprotinin) with or without 100 U/ml hexokinase (Type III from baker's yeast; Sigma) as described above and samples were loaded immediately on native or SDS polyacrylamide gels. Phosphatase inhibitors (10 mM tetrasodium pyrophosphate, 100 mM sodium fluoride, 17.5 mM β-glycerophosphate) were added to the lysis buffer in experiments where phosphorylated proteins were detected.

## Native polyacrylamide gel electrophoresis (native-PAGE)

Native-PAGE was performed as described (*Preissler et al., 2015*). Tris-glycine polyacrylamide gels consisting of a 4.5% stacking gel and a 7.5% separation gel were used to separate purified protein or proteins from mammalian cell lysates under non-denaturing conditions to detect BiP oligomers. The gels were run in a Mini-PROTEAN electrophoresis chamber (Bio-Rad) in running buffer (25 mM Tris, 192 mM glycine, pH ~8.8) at 120 V for 2 hr when cell lysates were applied or for 1:45 hr when His6-tagged purified BiP proteins were analyzed. The proteins were then visualized by staining with InstantBlue Coomassie solution (expedeon, UK) or transferred for immunodetection to a polyvinylidene difluoride (PVDF) membrane in blotting buffer (48 mM Tris, 39 mM glycine, pH ~9.2) containing 0.04 (w/v) SDS for 16 hr at 30 V. The membrane was washed after the transfer for 20 min in blotting buffer supplemented with 20% (v/v) methanol before blocking. Seven µg of purified BiP protein was loaded per lane on a native gel to detect BiP oligomers by Coomassie staining and volumes of lysates corresponding to 30 µg of protein (CHO-K1 and AR42j cells) or 90 µg protein (Flp-In T-REx 293 cells) were loaded per lane.

## Immunoblot analysis

Proteins were separated by SDS-PAGE or native-PAGE and transferred to PVDF membranes. The membranes were blocked with 5% (w/v) dried skimmed milk in TBS (25 mM Tris-HCl pH 7.5, 150 mM NaCl) and probed with primary antibodies followed by IRDye fluorescently labeled secondary antibodies (Li-Cor, UK). The membranes were scanned with an Odyssey near-infrared imager (Li-Cor) and where indicated densitometric quantification of the immunoblot signals was performed with ImageJ (NIH). Primary antibodies and antisera against hamster BiP [chicken anti-BiP; (*Avezov et al.,*

2013)], eIF2α [mouse anti-eIF2α; (Scorsone et al., 1987)], phosphorylated eIF2α [rabbit anti-eIF2α Phospho (pS51); Epitomics cat. # 1090-1, UK], and FICD [rabbit anti-FICD; LifeSpan BioSciences cat. # LS-C80941, UK or chicken anti-FICD (see below)] were used.

## Phos-tag gel electrophoresis

To separate unmodified IRE1α and phosphorylated IRE1α lysates from AR42j cells were loaded on SDS polyacrylamaide gels (Mini-PROTEAN system; Bio-Rad) consisting of a 4% polyacrylamide stacking gel and a 7% polyacrylamide separation gel. The separation gel contained 50 µM $MnCl_2$ and 25 µM Phos-tag Acrylamide AAL-107 (NARD Institute Ltd., Japan). The gels were run for 3 hr at 300 V, incubated for 10 min in immunoblot transfer buffer containing 1 mM EDTA and proteins were transferred to PVDF membranes.

## Protein purification

Wildtype and mutant versions of N-terminally hexahistidine- (His6-) tagged hamster BiP proteins (see *Supplementary file 1* and "Plasmid construction") were expressed in M15 *E. coli* cells (Qiagen, Germany) as described (*Preissler et al., 2015*). Bacterial cultures were grown at 37°C in LB medium containing 100 µg/ml ampicillin and 50 µg/ml kanamycin to an optical density ($OD_{600 nm}$) of 0.8 and expression was induced with 1 mM isopropylthio β-D-1-galactopyranoside (IPTG). After incubation for 6 hr at 37°C the cells were harvested by centrifugation and lysed with a high-pressure homogenizer (EmulsiFlex-C3; Avestin) in buffer A [50 mM Tris-HCl pH 7.5, 500 mM NaCl, 1 mM $MgCl_2$, 0.2% (v/v) Triton X-100, 10% (v/v) glycerol, 20 mM imidazole] containing protease inhibitors (2 mM PMSF, 4 µg/ml pepstatin, 4 µg/ml leupeptin, 8 µg/ml aprotinin) and 0.1 mg/ml DNaseI. The lysates were cleared by centrifugation (30 min at 25,000 *g*) and incubated with 1 ml nickel affinity matrix (Ni-NTA agarose; Qiagen) per 1 l of expression culture for 2 hr at 4°C. The beads were washed five times with 20 bed volumes of buffer A sequentially supplemented with (i) 30 mM imidazole, (ii) 1% (v/v) Triton X-100, (iii) 1 M NaCl, (iv) 5 mM $Mg^{2+}$-ATP, or (v) 0.5 M Tris-HCl pH 7.5. Bound BiP proteins were eluted in buffer B [50 mM Tris-HCl pH 7.5, 500 mM NaCl, 1 mM $MgCl_2$, 10% (v/v) glycerol, 250 mM imidazole] and dialyzed against HKM buffer (50 mM HEPES-KOH pH 7.4, 150 mM KCl, 10 mM $MgCl_2$). The purified proteins were concentrated using centrifugal filters (Amicon Ultra, 30 kDa MWCO; Merck Millipore, UK), snap-frozen in liquid nitrogen and stored at -80°C. The purification of mature hamster BiP (19-654) was described in detail in *Preissler et al., 2015*.

Mammalian expressed GST-tagged recombinant FICD[E234G] was produced in HEK293T cells. The cells were grown on 10 cm tissue culture dishes to a density of 80% and transfected either with plasmid encoding only GST or GST-FICD[E234G] (UK1415) using the polyethylenimine (PEI) method with the following adaptations: 14 µg plasmid DNA was added to 700 µl Opti-MEM (Gibco; Life Technologies, UK) mixed with 56 µl of PEI (1 mg/ml stock) and added dropwise to the cells after incubation for 20 min at room temperature (ten dishes were usually transfected per preparation). Thirty-six hr after transfection the cells were lysed as described above with Triton lysis buffer [20 mM HEPES-KOH pH 7.4, 150 mM NaCl, 1 mM EDTA, 10% (v/v) glycerol, 1% (v/v) Triton X-100] containing protease inhibitors. The combined lysate (from ten dishes) was incubated with 200 µl GSH-Sepharose beads (Glutathione Sepharose 4B; GE Healthcare, UK) for 3 hr at 4°C. The beads were recovered by centrifugation for 5 min at 500 *g*, washed twice with Triton lysis buffer containing 300 mM NaCl and twice with AMPylation buffer [25 mM HEPES-KOH pH 7.4, 100 mM KCl, 4 mM $MgCl_2$, 1 mM $CaCl_2$, 0.1% (v/v) Triton X-100]. The beads with bound protein were stored in AMPylation buffer containing 50% (v/v) glycerol at -20°C and used for radioactive in vitro AMPylation reactions.

For bacterial expression of N-terminally GST-tagged FICD proteins and ERdj6 J-domains, plasmid DNA encoding active GST-FICD[E234G] (UK1479), inactive GST-FICD[E234G-H363A] (UK1480), GST-J (UK185) or mutant GST-J* (UK186) was transformed into BL21 T7 Express *lysY/Iq E. coli* cells (New England BioLabs cat. # C3013, UK). Bacterial cultures were grown at 37°C in LB medium containing 100 µg/ml ampicillin to $OD_{600 nm}$ 0.8, shifted to 20°C and expression was induced with 0.1 mM IPTG for 16 hr. Afterwards, the cells were harvested and lysed as described above in lysis buffer [50 mM Tris-HCl pH 7.5, 500 mM NaCl, 1 mM $MgCl_2$, 2 mM dithiothreitol (DTT), 0.2% (v/v) Triton X-100, 10% (v/v) glycerol] containing protease inhibitors and DNaseI. The lysates were cleared by centrifugation (30 min at 25,000 *g*) and incubated with 0.7 ml GSH-Sepharose beads per 1 l of expression culture for 3 hr at 4°C. The beads were transferred to a column and washed with 20 ml wash buffer

B [50 mM Tris-HCl pH 7.5, 500 mM NaCl, 1 mM DTT, 0.2% (v/v) Triton X-100, 10% (v/v) glycerol] containing protease inhibitors, 20 ml wash buffer C [50 mM Tris-HCl pH 7.5, 300 mM NaCl, 10 mM MgCl$_2$, 1 mM DTT, 0.1% (v/v) Triton X-100, 10% (v/v) glycerol] containing protease inhibitors and 20 ml wash buffer C sequentially supplemented with (i) 1% (v/v) Triton X-100, (ii) 1 M NaCl, (iii) 3 mM ATP, (iv) or 0.5 M Tris-HCl pH 7.5. Bound BiP proteins were then eluted in elution buffer [50 mM HEPES-KOH pH 7.4, 100 mM KCl, 4 mM MgCl$_2$, 1 mM CaCl$_2$, 0.1% (v/v) Triton X-100, 10% (v/v) glycerol, 40 mM reduced glutathione] and concentrated protein solutions were frozen in liquid nitrogen and stored at -80°C.

N-terminally His6-Smt3 tagged wildtype mouse FICD lumenal domain (L104-P458) (UK1564) was encoded on a pET28b plasmid (Novagen, UK) and expressed in *E. coli* BL21 T7 Express *lysY/I$^q$* cells. Bacterial cultures (3 liters) were grown at 37°C in LB medium containing 100 µg/ml ampicillin to OD$_{600 nm}$ 0.8 and expression was induced with 0.1 mM IPTG. After incubation for 16 hr at 20°C the cells were sedimented by centrifugation and pellets were lysed as described above in lysis buffer [50 mM Tris-HCl pH 7.4, 500 mM NaCl, 1 mM MgCl$_2$, 10% (v/v) glycerol, 30 mM imidazole, 1 mM tris(2-carboxyethyl)phosphine (TCEP)] supplemented with protease inhibitors and DNaseI. The lysates were cleared by centrifugation for 30 min at 33,000 *g*, followed by addition of 0.2% (v/v) Triton X-100 and 0.002% (v/v) sodium deoxycholate, and incubated with 1 ml Ni-NTA agarose (Qiagen) for 2.5 hr at 4°C. The beads were washed with 25 bed volumes of wash buffer [50 mM Tris-HCl pH 7.4, 500 mM NaCl, 10% (v/v) glycerol, 0.2% (v/v) Triton X-100, 1 mM TCEP, protease inhibitors], once with buffer D [50 mM Tris-HCl pH 7.4, 300 mM NaCl, 10% (v/v) glycerol, 0.1% (v/v) Triton X-100, 10 mM MgCl$_2$, 1 mM TCEP, and protease inhibitors], and then with buffers E$_{i-iv}$ [buffer D supplemented with (i) 1% (v/v) Triton X-100, (ii) 1 M NaCl, (iii) 3 mM ATP, or (iv) 0.5 M Tris-HCl pH 7.4]. Bound mFICD protein was eluted in buffer F [50 mM Tris-HCl pH 7.4, 100 mM NaCl, 4 mM MgCl$_2$, 1 mM CaCl$_2$, 10% (v/v) glycerol, 250 mM imidazole, 1 mM TCEP), and dialyzed against 50 mM HEPES-NaOH pH 7.4, 100 mM NaCl, 0.5 mM 2-mercaptoethanol in the presence of Ulp1 (2 µg per mg of eluted protein) for 16 hr at 4°C to cleave the His6-Smt3 tag. The dialyzed and cleaved protein was then diluted 1:2 in 25 mM HEPES-NaOH pH 8.1 to reduce the ionic strength, loaded on a anion-exchange chromatography column (Mono Q 5/50 GL; GE Healthcare) and eluted with a linear salt gradient from 50 to 500 mM NaCl in 25 mM HEPES-NaOH pH 7.4. The mFICD-containing fractions were pooled and further purified by size-exclusion chromatography (Superdex 200 10/300; GE Healthcare) in gel filtration buffer (25 mM HEPES-NaOH pH 7.4, 150 mM NaCl). The eluted fractions were combined and provided to Aves Labs (Tigard, OR) as an immunogen for production of chicken anti-FICD antibodies.

## Immunoaffinity purification and immunoblotting of FICD

The absence of FICD protein in AR42j *FICD$^{-/-}$* cells was confirmed by immunoaffinity purification as the endogenous protein could not be detected by immunoblotting of total lysates. Protein extracts from wildtype and *FICD$^{-/-}$* AR42j cells were prepared in HG lysis buffer containing protease inhibitors, cleared, normalized and equal volumes of the lysates (4 mg total protein) were incubated with 15 µl UltraLink Hydrazine Resin (Pierce cat. # 53149, UK) on which FICD-specific chicken IgY antibodies have been covalently immobilized according to the manufacturer's instructions (chicken anti-FICD) or rabbit polyclonal IgG antibodies against human FICD (rabbit anti-FICD; LifeSpan BioSciences) bound to Protein A Sepharose 4B beads (Invitrogen cat. #10-1041), for 16 hr at 4°C. The beads were then recovered by centrifugation for 1 min at 1,000 *g* and washed three times with HG lysis buffer. Bound proteins were eluted in 40 µl 2 x SDS sample buffer for 10 min at 70°C and equal volumes of the samples were loaded on a 12% SDS polyacrylamide gel, and endogenous FICD was detected by immunoblotting with rabbit anti-FICD or chicken anti-FICD antibodies. Samples of the normalized lysates (25 µg) were loaded and BiP was detected as an "input" control.

## In vitro AMPylation assay

Unless indicated otherwise, in vitro AMPylation was performed in AMPylation buffer (described above) by incubation of purified wildtype or mutant BiP proteins at 20 µM with bacterially expressed GST-FICD$^{E234G}$ (or as a control with inactive GST-FICD$^{E234G-H363A}$) at 0.8 µM in presence of 1.5 mM ATP for 45 min at 30°C. The final volume of a typical reaction was 15 µl. The samples were then treated with or without the BiP linker-specific protease SubA at 30 ng/µl for 10 min at room

temperature. Volumes corresponding to 7 µg BiP protein were loaded per lane on native or SDS polyacrylamide gels and visualized by Coomassie staining. The in vitro AMPylation time-course experiments presented in *Figure 7C and D* were performed likewise but after incubation for the indicated times at 30°C the reactions were stopped on ice by addition of 40 mM EDTA and samples were run immediately on native gels.

Radioactive in vitro AMPylation was performed by mixing 7.5 µg of purified wildtype or mutant BiP proteins with 10 µM non-radiolabeled ATP and 0.185 MBq $\alpha$-$^{32}$P-ATP (EasyTide; Perkin Elmer, UK) in a total volume of 30 µl in AMPylation buffer. The reactions were incubated with 5 µl GSH-Sepharose beads coupled with mammalian expressed GST-FICD$^{E234G}$ (or GST only) in 1.5 ml reaction tubes at 30°C on a thermomixer (Eppendorf) while shaking at 800 rpm. After 45 min [in these experiments a time before the reaction went to completion to reveal differences in AMPylation amongst BiP mutants (*Figure 5A and C* and *Figure 7A*)] the beads were sedimented by centrifugation and the reaction was terminated by addition of SDS loading buffer to the supernatants and heating for 5 min at 75°C. Equal volumes of the samples were loaded on 12% SDS polyacrylamide gels. Proteins were visualized by Coomassie staining and radioactive signals were detected with a Typhoon Trio imager (GE Healthcare) upon overnight exposure of the dried gels to a storage phosphor screen. To test the sensitivity of AMPylated BiP towards cleavage by SubA (*Figure 3—figure supplement 1*), 6 µg of purified BiP$^{T229A}$ were labeled in 60 µl AMPylation buffer containing 6.7 MBq of $\alpha$-$^{32}$P-ATP and 10 µl of GST-FICD$^{E234G}$-coupled beads as described above. After incubation for 30 min at 30°C non-radioactive ATP was added at 5 µM and the reaction was allowed to proceed for another 60 min. The beads were then sedimented by centrifugation and 1.5 mM ATP was added to the supernatant. To remove nucleotides the sample was passed through a Sephadex G-50 Micro-Spin column (illustra AutoSeq G-50; GE Healthcare) equilibrated with AMPylation buffer. 1.5 µg of the recovered protein was mixed with 60 µg unmodified BiP$^{T229A}$ (at 0.6 mg/ml final) in presence of 1.5 mM ATP and the combined sample was divided into fractions to which SubA (from a dilution series) was added to yield final concentrations between 0.08 and 120 ng/µl. After incubation for 30 min at room temperature the proteins were denatured with SDS sample buffer and equal volumes were applied to SDS-PAGE, Coomassie-stained and signals were detected by autoradiography.

## Purification of in vitro AMPylated BiP proteins

Twenty mg of purified wiltype BiP or BiP$^{V461F}$ protein were in vitro AMPylated for 2 hr at 30°C with 1 mg bacterially expressed GST-FICD$^{E234G}$ in presence of 1.5 mM ATP in AMPylation buffer in a final volume of 12 ml. BiP proteins were then bound to 400 µl nickel affinity matrix, washed with AMPylation buffer and eluted in AMPylation buffer containing 250 mM imidazole. The concentrated eluate was divided into aliquots and immediately frozen in liquid nitrogen and stored at -80°C. For applications that required complete absence of detergent or imidazole the affinity chromatography eluate was passed through a Centri•Pure P25 desalting column (emp BIOTECH, Germany) equilibrated in HKM buffer followed by concentration of the proteins using centrifugal filters (Amicon Ultra, 30 kDa MWCO; Merck Millipore) before freezing. The in vitro modified proteins were used for mass spectrometry analyses and functional assays (see below). Unmodified BiP from parallel mock in vitro AMPylation reactions, to which no enzyme has been added, served as controls in all assays.

## Preparation of SILAC samples

The experimental strategy for the stable isotope labeling by amino acids in cell culture (SILAC) experiment is outlined in *Figure 6A* and samples were prepared as follows: Wildtype and *FICD$^{-/-}$* CHO-K1 cells were adapted to Ham's F12 medium minus L-arginine and L-lysine for SILAC (Pierce cat. # 88424) supplemented with 10% (v/v) dialyzed fetal bovine serum (Gibco; Life Technologies cat. # 26400-044), 1 x Penicillin-Streptomycin (Sigma), 2 mM L-glutamine (Sigma), 280 mg/l L-proline (Sigma cat. # P5607-25G), 62.5 mg/l "light" L-lysine monohydrochloride (Sigma cat. # L8662-25G) and 60.5 mg/l "light" L-arginine monohydrochloride (Sigma cat. # A6969-25G) and incubated as described above. Once adapted, the cells were cultured in SILAC medium containing either 60.5 mg/l "light" (as above) or "heavy" L-arginine monohydrochloride (R10; U-13C6; U-15N4; Cambridge Isotope Laboratories, Inc. cat. # CNLM-539, Andover, MO) for several passages (> 15 cell divisions) before expansion in five 10 cm dishes per sample. The cells were grown to ~70% confluence and medium was exchanged 14 hr before treatment with or without 100 µg/ml cycloheximide for 3 hr.

After washing with ice-cold PBS the cells were collected in PBS containing 1 mM EDTA and lysed in Triton lysis buffer [20 mM HEPES-KOH pH 7.4, 150 mM NaCl, 1 mM EDTA, 1% (v/v) Triton X-100, 10% (v/v) glycerol] containing protease inhibitors. The protein concentrations of the cleared lysates were normalized and equal volumes were then mixed (1:1 ratio) as indicated. Each combined sample was pre-cleared for 15 min at 4°C with 30 µl blocked UltraLink Hydrazine Resin followed by immunoaffinity purification for 2 hr at 4°C with 30 µl of the same resin to which BiP-specific IgY antibodies were covalently coupled (*Preissler et al., 2015*). The beads were washed twice for 10 min with lysis buffer, once with lysis buffer containing 350 mM NaCl and once again with lysis buffer, before elution in 2x non-reducing SDS sample buffer for 10 min at 70°C. Afterwards, the beads were removed by centrifugation and the supernatants were supplemented with 125 mM DTT, heated for 5 min at 70°C and loaded on a 12% SDS polyacrylamide gel. The gel was stained with Coomassie and bands corresponding in size to BiP were cut out for in-gel digest with Arg-C endopeptidase, and analysis by mass spectrometry.

## Immunoaffinity purification of intact endogenous BiP for mass determination

Wildtype and *FICD*[-/-] CHO-K1 cells were grown in standard medium (ten 10 cm dishes per sample) and treated for 3 hr with 100 µg/ml cycloheximide before cell lysis with Triton lysis buffer supplemented with protease inhibitors and BiP was immunoaffinity purified as described above with the following modifications: The lysates (~4 mg protein per sample) were pre-cleared with 40 µl blocked UltraLink Hydrazine Resin for 30 min at 4°C followed by incubation for 2 hr with 60 µl of the resin carrying covalently attached BiP-specific IgY antibodies. The beads were washed sequentially for 10 min with lysis buffer, lysis buffer containing 350 mM NaCl, HG buffer (see above) containing 1 mM ATP, and once again with lysis buffer. Bound protein was eluted in 150 µl of 0.2 M glycine-HCl pH 2.5 for 10 min at 16°C followed by neutralization of the recovered sample supernatants with 1/10 volume of 1.5 M Tris-HCl pH 8.8. Afterwards, protein was precipitated with isopropanol and analyzed by mass spectrometry.

## Mass spectrometry

Molecular masses of intact BiP proteins were measured by electrospray ionization mass spectrometry using a Q-Trap 4000 (ABSciex) after 'on-line' reverse-phase chromatographic purification of samples as described previously (*Carroll et al., 2009*). The instrument was operated in MS mode, and was calibrated with a mixture of myoglobin and trypsinogen. Reconstructed molecular masses were calculated with Bioanalyst (ABSciex) and masses from amino acid sequences with MassLynx (Waters).

To analyze masses of BiP peptides, proteins were reduced, alkylated and digested "in-gel". In vitro AMPylated BiP protein was digested with the proteolytic enzymes as indicated and SILAC samples were digested using Arg-C. The resulting peptides were analyzed by LC-MS/MS using either an Orbitrap XL coupled to a nanoAcquity UHPLC or a Q Exactive coupled to a RSLC3000 UHPLC. Data was processed in Proteome Discoverer 1.4 using Sequest to search a UniProt *E. coli* database (downloaded 29/04/15, 4,378 entries) with the sequence for Chinese hamster (*Cricetulus griseus*) BiP added or a Chinese Hamster database (downloaded 19/02/14, 23,884 entries). Oxidation (M), deamidation (N/Q) and AMPylation (S/T) were set as variable modifications and carbamidomethylation (C) as a fixed modification. FDR calculations were performed by Percolator and peptides were filtered to 1%. Peak areas were determined using the Precursor Ion Quantifier node in Proteome Discoverer.

## Isoelectric focusing (IEF)

Purified untagged hamster BiP[19-654] at 15 µM was in vitro AMPylated with 0.75 µM GST-FICD[E234G] (or as a control with inactive GST-FICD[E234G-H363A]) in presence of 1.5 mM ATP for the indicated times at 30°C as described above. The reactions were then treated without or with 30 ng/µl SubA for 10 min at room temperature in a final volume of 20 µl and analyzed by IEF as described previously (*Chambers et al., 2012*; *Laitusis et al., 1999*) with modifications: The samples were diluted with 180 µl of IEF sample buffer [8 M urea, 5% (w/v) CHAPS, 50 mM DTT, 2% (v/v) Pharmalyte (pH 4.5-5.4; GE Healthcare)] and loaded on a 3.75% polyacrylamide gel containing 8.8 M urea, 1.25% (w/v) CHAPS and 5% (v/v) Pharmalyte. The loaded samples were overlaid with 0.5 M urea and 2% (v/

v) Pharmalyte solution before separation. 10 mM glutamic acid and 50 mM histidine were used as anode and cathode buffers, respectively. After the run (100V, 10 min; 250V, 1 hr; 300V, 1 hr; 500V, 30 min) proteins were transferred to a nitrocellulose membrane in blotting buffer [25 mM Tris-HCl pH 9.2, 190 mM glycine, 0.01% (w/v) SDS, 10% (v/v) methanol] for 3 hr at 300 mA and immunodetected with BiP-specific antibodies as described above.

Lysate samples from mammalian cells (CHO-K1 and AR42j) were prepared for analysis by IEF as follows: Cells were grown in 10 cm dishes to approximately 90% confluence, washed with ice-cold TBS, resuspended in 1 ml TBS, sedimented by centrifugation, and lysed in 30 x its packed cell volume of IEF lysis buffer [8.8 M urea, 5% (w/v) CHAPS, 1 µM sodium pyrophosphate, 2 mM imidodiphosphate, 50 mM DTT and 2% (v/v) Pharmalyte] at room temperature for 5 min. The lysates were then centrifuged at 20,238 $g$ for 10 min at room temperature, the supernatants were transferred to a new tube and then centrifuged again for 60 min. The cleared lysates (50 µl) were then purified using Sephadex G-50 MicroSpin columns equilibrated with IEF sample buffer, diluted 1:2 in IEF sample buffer and 15 µl were loaded on a 3.75% polyacrylamide gel followed by BiP immunoblotting as described above.

## Malachite green ATPase assay

Malachite green (MG) reaction solution was prepared freshly by mixing stock solutions of MG dye [0.111% (w/v) malachite green, 6 N sulphuric acid], 7.5% (w/v) ammonium molybdate and 11% (v/v) Tween 20 in a 10:2.5:0.2 ratio. Samples of purified BiP proteins (5 µM) were incubated in HKM with 3 mM ATP in a final volume of 20 µl for 60 min at 30°C. 15 µl of each sample were then diluted with 135 µl water on a 96-well plate, mixed with 50 µl MG reaction solution and incubated for 2 min at room temperature. 20 µl of a 34% (w/v) sodium citrate solution were added to quench the reactions and after incubation for further 30 min the absorbance was measured at 612 nm with a plate reader (Infinite F500; Tecan). Standard curves were prepared with serial dilutions of $KH_2PO_4$ and served as a reference to calculate $V_{max}$ values. Statistical analysis by unpaired t-test was performed using Graphpad Prism version 6.

## Fluorescence polarization (FP) measurements

Binding of substrate peptide by purified BiP proteins was measured as previously described (*Chambers et al., 2012*; *Marcinowski et al., 2011*) with modifications: Unmodified or in vitro AMPylated BiP proteins were incubated at the indicated concentrations (between 0 and 40 µM) with 1 µM lucifer yellow (LY)-labeled substrate peptide (HTFPAVLGSC) in presence of 1 mM ADP for 24 hr at 30°C in FP buffer [37.5 mM HEPES-KOH pH 7.4, 125 mM KCl, 7 mM $MgCl_2$, 0.5 mM $CaCl_2$, 125 mM imidazole, 0.05% (v/v) Triton X-100]. 20 µl of each sample were transferred to a 386-well polystyrene microplate (µClear, black; greiner bio-one cat. # 781096, UK) and fluorescence polarization of the lucifer yellow fluorophore was measured (excitation λ = 430 nm; emission λ = 535 nm) at room temperature in a plate reader. For analysis of competitive substrate dissociation, BiP proteins (40 µM) were incubated with LY-labeled substrate peptide (1 µM) as described above. Afterwards, 400-fold molar excess of unlabeled substrate peptide (HTFPAVL) was added to the pre-formed BiP:substrate complexes and the change in fluorescence polarization was measured over time. From each value the background signal of a reference sample containing only LY-labeled peptide (without BiP protein) were subtracted.

## Reverse transcription of cellular mRNA and quantitative PCR

RNA from CHO-K1 cells was extracted using RNA Stat-60 (Amsbio, UK) according to the manufacturer's instructions. 1 µg RNA was reverse transcribed with Oligo(dT)18 primer using RevertAid Reverse Transcriptase (Thermo Scientific, UK). Quantitative PCR analysis was performed using the Power SYBR Green PCR Master Mix (Applied Biosystems) according to the manufacturer's instructions on a 7900HT Fast Real-Time PCR System (Applied Biosystems). *FICD, RPL27* (60S ribosomal protein L27) and *PPIA* (cyclophilin A) were amplified using the following primers: FICD-F 5'-GGGCT-GCTGACTGTTAAGACTA-3', FICD-R 5'-CCTTGACGCTTCATCTCCA-3', RPL27-F 5'-ACAATCACCT-CATGCCCACAAG-3', RPL27-R 5'-GCGTTTCAGGGCTGGGTCTC-3', PPIA-F 5'-TTCCTCCTTTCACAGAATTATCCC-3', and PPIA-R 5'-CTGCCGCCAGTGCCATTATG-3'. Relative

quantities of amplified cDNAs were then determined using SDS 2.4.1 software (Applied Biosystems) and normalized to *PPIA* mRNA.

## Acknowledgements

We thank the CIMR flow cytometry core facility team (Reiner Schulte, Michal Maj and Chiara Cossetti) for assistance with FACS, Heather P. Harding, Niko Amin-Wetzel and Joseph E. Chambers (all from CIMR, Cambridge) for technical advice and critical input, Qinlian Liu from Virginia Commonwealth University for sharing unpublished structural data, and Claudia Flandoli (www.claudiaflandoli.com) for preparation of the cartoon images.

Supported by grants from the Wellcome Trust (Wellcome 084812/Z/08/Z) the European Commission (EU FP7 Beta-Bat No: 277713), a Wellcome Trust Strategic Award for core facilities to the Cambridge Institute for Medical Research (Wellcome 100140). DR is a Wellcome Trust Principal Research Fellow.

## Additional information

### Competing interests

DR: Reviewing editor, *eLife.* The other authors declare that no competing interests exist.

### Funding

| Funder | Grant reference number | Author |
| --- | --- | --- |
| Wellcome Trust | Wellcome 084812/Z/08/Z | David Ron |
| European Commission | EU FP7 Beta-Bat No: 277713 | David Ron |
| Medical Research Council | | Ian M Fearnley |

The funders had no role in study design, data collection and interpretation, or the decision to submit the work for publication.

### Author contributions

SP, Co-led the project, designed and interpreted the bulk of the experiments, prepared mass spectrometry samples, performed biochemical in vitro experiments, contributed to in vivo experiments and wrote the paper; CR, Co-led the project, created the cell lines, designed and conducted and interpreted the bulk of the in vivo experiments and contributed to the writing and editing of the manuscript; RC, Contributed to the analysis of BiP AMPylation in vitro; RA, Mass spectrometry measurements and analysis; SD, Mass spectrometry measurements and analysis of intact BiP; IMF, Mass spectrometry measurements, analysis of intact BiP and editing the manuscript; DR, Conceived and oversaw the study as a whole, wrote the paper and designed and constructed expression plasmids for the study.

### Author ORCIDs

Steffen Preissler, http://orcid.org/0000-0001-7936-9836
David Ron, http://orcid.org/0000-0002-3014-5636

## Additional files

### Supplementary files

• Supplementary File 1. List of plasmids used.

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
