## [Decision Letter]

Thank you for submitting your work entitled "AMPylation matches BiP activity to client protein load to prevent over-chaperoning of the endoplasmic reticulum" for consideration by *eLife*. Your article has been favorably evaluated by Randy Schekman (Senior Editor) and three reviewers, one of whom, Reid Gilmore, is a member of our Board of Reviewing Editors.

The reviewers have discussed the reviews with one another and the Reviewing Editor has drafted this decision to help you prepare a revised submission.

Summary:

The manuscript from Preissler et al. investigates AMPylation of BiP. BiP was known to undergo a modification in the ER, which was believed to be ADP-ribosylation based upon incorporation of phosphate and adenosine. Previously, the Ron lab indirectly mapped the sites of ADP-ribosylation to two arginines in the substrate-binding domain (SBD) of BiP. Two recent papers established that the modification is AMPylation, not ADP-ribosylation, and identified an ER localized enzyme (FICD/HYPE) that is likely responsible for the modification. However, contradictory conclusions were reached concerning the location of the AMPylated residue (ATPase domain or SBD) and the impact of AMPylation on BiP activity and UPR signaling. In the current manuscript, Preissler et al. use a combination of in vitro AMPylation experiments and cell culture experiments to convincingly demonstrate the following: (a) FICD modifies a single site in the substrate binding domain of BiP both in vitro and in cells, (b) the primary modification site is T518, (c) AMPylation of BiP reduces BiPs ATPase rate and renders it less-responsive to J domain containing proteins, (d) excessive AMPylation of BiP is toxic to cells due to constitutive activation of the UPR and (e) a block in AMPylation of BiP blunts the early response of cells to ER stress. One weakness of the paper is that the effects of constitutive AMPylation or inhibited AMPylation on UPR signaling are relatively small. It thus remains unclear whether the modification has an important physiological role. The subtle change in the kinetics of UPR activation that was observed in the FIDC null cells is probably not sufficient to warrant the title of the manuscript. This is a solid and interesting paper that should be appropriate for publication in *eLife* after modifications to address the major points listed below.

Essential revisions:

1) The manuscript should include a serious discussion of the previous data reported by the Ron group and other labs to clarify whether a modification at T518 can account for the previous findings. For example, is the weaker effect of point mutations (R470E and R492E) on in vitro AMPylation in the current manuscript explained by differences in assay conditions? The Ham and Orth groups reported AMPylation in the NBD, yet reached conflicting conclusions on the effect of this modification on ATP hydrolysis by BiP. The authors should comment on how a modification of a threonine residue in a loop linking the SBD and how NBD could affect ATP hydrolysis.

2) The other two groups reported that FicD/HYPE is up-regulated by ER stress. Have the authors of the current manuscript determined whether HYPE is induced by the UPR? If so, are the kinetics of up-regulation slow enough to suggest that UPR induced HYPE expression is a mechanism to inactivate the excess BiP made during a stress response? If not how does this fit their hypothesis?

3) A central conclusion of the current manuscript is that modification of a single site (T518) can account for all previous evidence that BiP was modified by adenosine and phosphate. The BiP and AMPylated BiP that were used for electrospray mass spectroscopy eluted as broad, tailing peaks on the LC column (Figure 4—figure supplement 1). Prominent shoulders (71.3 min and 73.7 min) were present in both the BiP and BiP-AMP samples. Were the tailing sections of the BiP and BiP-AMP peak analyzed by ES-MS, and do they contain BiP with different, or additional modifications?

4) Why is the FL/BiP ratio high (7.18, 7.48) in cells that have the constitutively active FICD plasmid in the absence of doxycycline (Figure 9)? Are low levels of FICD expressed in the absence of DOX? Faint bands in -Dox lanes also seem to be present in control cells, which do not have a high FL/BiP ratio. The authors should comment on this discrepancy.

5) The kinetics of peptide dissociation from BiP-AMP looks very biphasic even when the time scale is 0-60 min. The data should also be fit assuming a biphasic dissociation rate. If such a fit is superior, this raises the question of whether the BiP-AMP sample is homogeneous. Could it contain some unmodified BiP? Or could it contain BiP AMPylated at a different site (T525 or T527)? The figure could be improved by having a dashed line showing the fluorescence polarization of free peptide expressed relative to the BiP-peptide and BiP-AMP peptide complexes. Figure 8 is the best evidence that AMPylation of BiP has an effect on the behavior of the chaperone in vitro, so the data need to be convincing. Please provide error bars.

6) The authors present evidence that the ATPase activity of BiP-AMP is not stimulated by the J-domain of ERdJ6. Failure of a J domain to stimulate hydrolysis of the modified protein is somewhat surprising, particularly since they have concluded that the ATP bound form of BiP is the substrate for AMPylation, which is also the preferred binding form for DnaJ proteins. Does the J domain still bind ATP BiP-AMP? Or do they suppose the AMPylation stimulate ATP hydrolysis rendering BiP resistant to J domain stimulation?

---

## [Author Response]

Essential revisions:

1) The manuscript should include a serious discussion of the previous data reported by the Ron group and other labs to clarify whether a modification at T518 can account for the previous findings. For example, is the weaker effect of point mutations (R470E and R492E) on in vitro AMPylation in the current manuscript explained by differences in assay conditions? The Ham and Orth groups reported AMPylation in the NBD, yet reached conflicting conclusions on the effect of this modification on ATP hydrolysis by BiP. The authors should comment on how a modification of a threonine residue in a loop linking the SBD and how NBD could affect ATP hydrolysis.

A new paragraph in the Discussion is now devoted to the first part of this critique:

“An earlier study from our lab revealed that mutations affecting residues R470 and R492 consistently diminished labeling of over-expressed mutant BiP from ^32^P orthophosphate or ^3^H-adenosine pools in vivo(Chambers et al., 2012), leading us to suggest that these residues undergo ADP-ribosylation. […] Such defects appear to have a more conspicuous effect on modification of over-expressed BiPin vivo thanon the modification of pure recombinant BiP by FICD in vitro.”

The second point in the critique above is now addressed in the preceding paragraph in the Discussion:

“Thr^518^ of hamster BiP is highly conserved in other eukaryotes. […] It seems plausible that a modification affecting the disposition of this loop could reach the NBD through a reciprocal set of allosteric interactions to explain the effects of AMPylation on nucleotide hydrolysis and responsiveness to J proteins.”

*2) The other two groups reported that FicD/HYPE is up-regulated by ER stress. Have the authors of the current manuscript determined whether HYPE is induced by the UPR? If so, are the kinetics of up-regulation slow enough to suggest that UPR induced HYPE expression is a mechanism to inactivate the excess BiP made during a stress response? If not how does this fit their hypothesis?*

New Figure 2—figure supplement 1 addresses this critique. It shows a progressive increase in the fraction of BiP susceptible to SubA cleavage and a correlated disappearance of the ‘B’ form on native-PAGE observed in wildtype cells following exposure to the reversible ER stress-inducing agent 2-deoxy-D-glucose (2-DG) and a correlated re-emergence of ‘B’ form and SubA-resistant BiP during the 2-DG washout; neither of which are evident in the *FICD^-/-^* cells. Declining levels of modified BiP are observed during mounting stress and are dissociated from the previously-reported increase in *FICD* mRNA (which we too observe, panel D). These findings support tight regulation of FICD enzymatic activity and are discussed in detail in the Results section and the Discussion.

In regards to the specific comment in the second half of the critique we note that our findings are consistent with a role to transcriptional induction of FICD as playing a role in the recovery of modified BiP during the resolution of the stress. However, the dissociation between mounting levels of *FICD* mRNA and declining/low levels of modified BiP in the first 8 hr of a the stress response suggest that the dominant regulatory circuitry is not played out at the level of transcriptional regulation of *FICD*.

*3) A central conclusion of the current manuscript is that modification of a single site (T518) can account for all previous evidence that BiP was modified by adenosine and phosphate. The BiP and AMPylated BiP that were used for electrospray mass spectroscopy eluted as broad, tailing peaks on the LC column (Figure 4—figure supplement 1). Prominent shoulders (71.3 min and 73.7 min) were present in both the BiP and BiP-AMP samples. Were the tailing sections of the BiP and BiP-AMP peak analyzed by ES-MS, and do they contain BiP with different, or additional modifications?*

We have revisited the MS data and confirmed that the ion current in the trailing shoulder of BiP-containing peak is comprised of singly charged low molecular weight (likely non-proteinaceous) species and therefore unlikely to reflect heterogeneity in BiP modification.

In the revised paper we have now evaluated the native mass of endogenous BiP from wildtype and *FICD^-/-^* cells (new Figure 4—figure supplement 2). The mass spectrum is remarkable in its simplicity: *FICD^-/-^* cells have a single species of 70,532 Da whereas wildtype cells (exposed to cycloheximide to promote BiP AMPylation) have two species, 70,538 and 70,866 Da. These observations point to a single FICD-dependent AMPylation event as the only conspicuous modification of BiP in cultured CHO-K1 cells.

*4) Why is the FL/BiP ratio high (7.18, 7.48) in cells that have the constitutively active FICD plasmid in the absence of doxycycline (Figure 9)? Are low levels of FICD expressed in the absence of DOX? Faint bands in -Dox lanes also seem to be present in control cells, which do not have a high FL/BiP ratio. The authors should comment on this discrepancy.*

As noted in the Results section of the revised manuscript, “…the conspicuous basal (Dox-independent) SubA resistance of endogenous BiP, lanes 5 & 7, likely reflecting basal leakiness of the expression system”. We thank the reviewer(s) for flagging this issue.

*5) The kinetics of peptide dissociation from BiP-AMP looks very biphasic even when the time scale is 0-60 min. The data should also be fit assuming a biphasic dissociation rate. If such a fit is superior, this raises the question of whether the BiP-AMP sample is homogeneous. Could it contain some unmodified BiP? Or could it contain BiP AMPylated at a different site (T525 or T527)? The figure could be improved by having a dashed line showing the fluorescence polarization of free peptide expressed relative to the BiP-peptide and BiP-AMP peptide complexes. Figure 8 is the best evidence that AMPylation of BiP has an effect on the behavior of the chaperone in vitro, so the data need to be convincing. Please provide error bars.*

We have re-analyzed the data from all three experiments and found that they fit better to a mono-phasic than bi-phasic plot.

We now provide the source data on all three experiments, which includes the time-dependent changes in the fluorescent polarization signal arising from the fluorescently-labeled peptide alone.

In regards to the concern that recombinant BiP is heterogeneously AMPylated on the Arg-C BiP^511-532^ peptide, Figure 6—figure supplement 2 of the revised manuscript presents the MS/MS fragmentation data showing that the series of *y*-ions and *b*-ions derived from the fragmentation of unmodified BiP extend to all the residues of the parent peptide. However, ions containing Thr^518^ are conspicuously missing from the fragmentation spectra of modified BiP. This finding pin-points the modification to Thr^518^ and excludes Thr^525^ and Thr^527^.

6) The authors present evidence that the ATPase activity of BiP-AMP is not stimulated by the J-domain of ERdJ6. Failure of a J domain to stimulate hydrolysis of the modified protein is somewhat surprising, particularly since they have concluded that the ATP bound form of BiP is the substrate for AMPylation, which is also the preferred binding form for DnaJ proteins. Does the J domain still bind ATP BiP-AMP? Or do they suppose the AMPylation stimulate ATP hydrolysis rendering BiP resistant to J domain stimulation?

We agree with the reviewer(s) that the basis for the inhibition of responsiveness to J-protein by AMPylated BiP is one of the most interesting issues for future study, but at present we can only speculate that it too is a reflection of the reciprocal allostery from the SBD to the NBD (discussed above in response to critique #1). In defense of the plausibility of this idea we note the precedence provided by the Gly^461^Pro/Gly^468^Pro human BiP double mutant which, though locked in an ATP-like conformation, is nonetheless unresponsive to J-protein (Yang et al., 2015) (cited in the Discussion, fifth paragraph).

In regards to the very sensible suggestion by the reviewer(s) that we examine the effect of AMPylation on the interaction of BiP with DnaJ, we should like to point out that this issue is not easy to access experimentally: Whilst wildtype BiP (or other Hsp70’s) will avidly associate with immobilised J-proteins, in an ATP and functional J-domain dependent manner, forming a stable complex, this complex is not a good read-out of the BiP-J interaction the reviewers (and ourselves) would like measure. Its dependence on the ability of BiP to engage substrate in its SDB indicates that the interactions reflect the ability of the J protein to stimulate ATP hydrolysis driving BiP hyper affinity for substrates whilst serve as an unfolded substrate at the same time (see Figure Supplement 6 in Petrova et al. 2008; PMID: 18923430). We are actively seeking a more refined assay that gets around this problem, but are unable to offer one at present.